# Immuno-metabolic stress responses control longevity from mitochondrial translation inhibition in *C. elegans*

Iman Man Hu [1,2], Marte Molenaars[3], Yorrick R. J. Jaspers [1,2], Bauke V. Schomakers[1,2,4], Michel van Weeghel [1,2,4], Amber Bakker[1], Melanie Modder [1], Joseph P. Dewulf [5], Guido T. Bommer [5], Arwen W. Gao [1,2], Georges E. Janssens[1,2] & Riekelt H. Houtkooper [1,2,6] ✉

Perturbing mitochondrial translation represents a conserved longevity intervention, with proteostasis processes proposed to mediate the resulting lifespan extension. Here, we explore whether other mechanisms may contribute to lifespan extension upon mitochondrial translation inhibition. Using multiomics and functional in vivo screening, we identify the ethylmalonyl-CoA decarboxylase orthologue *C32E8.9* in *C. elegans* as an essential factor for longevity induced by mitochondrial translation inhibition. Reducing *C32E8.9* completely abolishes lifespan extension from mitochondrial translation inhibition, while mitochondrial unfolded protein response activation remains unaffected. We show that *C32E8.9* mediates immune responses and lipid remodeling, which play crucial roles in the observed lifespan extension. Mechanistically, *sma-4* (a TGF-β co-transcription factor) serves as an effector of *C32E8.9*, responsible for the immune response triggered by mitochondrial translation inhibition. Collectively, these findings underline the importance of the "immuno-metabolic stress responses" in longevity upon mitochondrial translation inhibition and identify *C32E8.9* as a central factor orchestrating these responses.

Mitochondria drive energy production and hold central positions in essential biological processes, including metabolism, apoptosis, immunity, and lifespan[1]. Research across vertebrates and invertebrates highlights the intricate association between mitochondrial biology and longevity[2]. With age, mitochondria undergo a progressive deterioration, manifesting as changes in mitochondrial morphology, abundance, and OXPHOS activity[3,4]. Conversely, mild inhibition of mitochondrial activity initiates protective responses that counteract age-dependent deterioration, at least in *C. elegans* and *D. melanogaster*[5–9].

A crucial adaptive stress response activated by mitochondrial stress is the mitochondrial unfolded protein response (UPR^mt)[10]. It is activated to preserve mitochondrial proteostasis by inducing the mitochondrial protein quality control machinery, which is regulated by the transcription factor Activating Transcription Factor associated with Stress-1 (ATFS-1)[11]. UPR^mt promotes repair and recovery from mitochondrial dysfunction by inducing the expression of cytoprotective genes[12,13]. We and others have previously shown that moderate mitochondrial translation inhibition through RNA interference (RNAi)

[1]Laboratory Genetic Metabolic Diseases, Amsterdam UMC Location University of Amsterdam, Amsterdam, The Netherlands. [2]Amsterdam Gastroenterology Endocrinology and Metabolism Institute, Amsterdam, The Netherlands. [3]Department of Pathology, New York University Grossman School of Medicine, New York, NY, USA. [4]Core Facility Metabolomics, Amsterdam UMC Location University of Amsterdam, Amsterdam, The Netherlands. [5]Department of Biochemistry, de Duve Institute, UCLouvain, Brussels, Belgium. [6]Amsterdam Cardiovascular Sciences Institute, Amsterdam, The Netherlands. ✉e-mail: r.h.houtkooper@amsterdamumc.nl

targeting mitochondrial ribosomal protein 5 (*mrps-5* in *C. elegans*) or pharmacologically by using doxycycline leads to lifespan extension and yields favorable outcomes in ameliorating disease models associated with aging, such as Alzheimer's and Parkinson's[5,6,14]. The prolonged lifespan resulting from mitochondrial translation repression is blunted by the inactivation of key components of the UPR[mt] [6,8], suggesting that these observed effects arise primarily due to UPR[mt] activation.

Despite the clear connection between UPR[mt] activation and longevity in the context of mitochondrial translation inhibition, UPR[mt] activation is also observed in short-lived mitochondrial mutants, and constitutively active alleles of ATFS-1 exhibit reduced lifespan[15]. These observations indicate that the relationship between UPR[mt] and longevity is not universal. We hence explored the molecular changes induced by slowing down mitochondrial translation, and uncovered that a conserved program for mitochondrial-cytosolic translational balance is activated upon repression of mitochondrial translation[16]. Indeed, the energy-costly process of cytosolic mRNA translation is shut down in response to mitochondrial translation inhibition, both in *C. elegans* and mammalian models[16]. This mitochondrial-cytosolic translational communication is dependent on the transcription factor *atf-4* (previously named *atf-5* in *C. elegans*, mammalian ATF4), which is required to assure longevity via mitochondrial translational inhibition in *C. elegans*. Yet, knockdown of *atf-4* can only partially reverse the lifespan extension by mitochondrial translational inhibition, suggesting that the downstream mechanisms for the beneficial effect of mitochondrial translation distress are not fully grasped. In this work, we identify a conserved set of genes activated by mitochondrial translation inhibition and establish *C32E8.9* as essential for the resulting healthspan and lifespan extension in *C. elegans*. This pathway does not involve the activation of the UPR[mt] and instead depends on activation of the immune response. We show that *C32E8.9* regulates both immune signaling and lipid remodeling, which are critical for the longevity effect. Our findings highlight an immuno-metabolic response that underlies the benefits of mitochondrial translation inhibition.

## Results

### *C32E8.9* drives beneficial effects mediated by mitochondrial translation inhibition

To identify conserved molecular signatures underlying the lifespan extension upon mitochondrial translation inhibition, we analyzed the transcriptome of doxycycline-treated germ-free mice and the transcriptome/proteome of *mrps-5* RNAi worms generated from our previous study[16] (Fig. 1a). We found 23 genes that overlap in expression between *C. elegans* and mice at both mRNA and protein levels. Among these genes, 19 upregulated genes serve as candidates contributing to longevity induced by mitochondrial translation inhibition (Fig. 1b). To identify which candidate genes were potentially responsible for the mitochondrial translation inhibition-induced longevity effect, we performed a systematic candidate gene approach using RNAi inhibition of the candidates. Specifically, we aimed to determine if knockdown of our candidate genes would blunt the improved healthspan of *mrps-5* RNAi in GMC101 worms. GMC101 is *C. elegans* model for Alzheimer's disease and is commonly used in screenings for potential therapeutics for age-related diseases[14]. These worms are engineered to express full-length human amyloid-beta (Aβ1-42) in body wall muscle cells and show an age-dependent paralysis phenotype when the cultured temperature is shifted from 20 to 25 °C[17]. Mobility assays showed that *mrps-5* RNAi enhanced the mobility of GMC101 maintained at 25 °C (Fig. 1c, d, Supplementary Fig. 1a). Thirteen out of 19 candidate genes were available in our RNAi library and were evaluated in GMC101 worms by performing healthspan assays. Briefly, we measured the mobility of GMC101 worms in four conditions: (1) control, (2) single knockdown of the candidate gene, (3) single knockdown of *mrps-5* and (4) double RNAi targeting the candidate gene alongside *mrps-5*. The screening revealed that knockdown of *gcs-1*, *dnj-13* and *C32E8.9*

significantly blunted the mobility improved by *mrps-5* RNAi (Fig. 1c), providing a basis for our second-tier screening.

*gcs-1* is part of the glutamate-cysteine ligase complex, which has been implicated in oxidative stress responses and lifespan modulation[18]. *dnj-13* is one of the HSP40-like J domain proteins that promote the clearance of protein aggregates[19]. We fed worms with *dnj-13* RNAi bacteria and observed higher mortality than the controls, suggesting it might be essential for development and survival (Supplementary Fig. 1a). Doxycycline is a commonly used pharmaceutical agent to inhibit mitochondrial translation. qPCR analysis revealed that the mRNA expression of *C32E8.9* was also upregulated following doxycycline treatment as well (Supplementary Fig. 1g). To investigate the role of *C32E8.9* under this condition, we assessed the mobility of GMC101 worms treated with doxycycline in combination with *C32E8.9* RNAi (Supplementary Fig. 1h). The results showed that knockdown of *C32E8.9* significantly compromised the improved mobility induced by doxycycline treatment, consistent with our observations under mitochondrial translation inhibition induced by *mrps-5* RNAi. *C32E8.9* is an ortholog of human *ECHDC1* (ethylmalonyl-CoA decarboxylase 1), which plays an important role in preventing the formation of abnormal fatty acids[20]. So far, there is no existing link between *C32E8.9* and lifespan regulation. As such, we sought to elucidate the role of *C32E8.9* in modulating the improved lifespan and healthspan upon mitochondrial translation inhibition. We fed worms with *C32E8.9* RNAi bacteria either alone or in combination with *mrps-5* RNAi. Subsequent qPCR analysis confirmed the knockdown efficiency. The expression of *C32E8.9* was reduced by 53% in worms fed solely with *C32E8.9* RNAi, while its expression was increased by 25% in worms treated with *mrps-5* RNAi (Supplementary Fig. 1b). To ensure that knockdown efficiency was maintained in the double RNAi experiments, we conducted qPCR analyses to measure the mRNA expression levels of both *mrps-5* and *C32E8.9*. We compared their expression in worms treated with single RNAi versus double RNAi conditions. For *mrps-5*, no significant difference in mRNA levels was detected between the single and double RNAi setups, confirming consistent knockdown efficiency (Supplementary Fig. 1d). Similarly, *C32E8.9* expression remained unaffected in the double RNAi condition (Supplementary Fig. 1c). The healthspan and lifespan showed that *C32E8.9* RNAi had minimal impact on either parameter when administered alone (Fig. 1d, e). However, *C32E8.9* RNAi completely abolished the enhanced healthspan and lifespan extension induced by *mrps-5* RNAi (Fig. 1d, e). To avoid potential off-target effects and incomplete knockdown, we generated a *C32E8.9* knockout strain using CRISPR/Cas9 technology. This knockout strain was successfully constructed and demonstrated to be both viable and fertile under laboratory conditions. qPCR analysis further verified the complete loss of *C32E8.9* expression in the mutant strain, as shown in Supplementary Fig. 1f. The lifespan assays revealed that the *C32E8.9* knockout produced effects similar to those observed with RNAi-mediated knockdown. Specifically, our results show that the lifespan extension induced by *mrps-5* RNAi was completely abolished in the *C32E8.9* knockout strain, confirming that *C32E8.9* is indeed necessary for this beneficial effect. Notably, the lifespan of the *C32E8.9* knockout strain itself did not significantly differ from that of wild-type controls (Fig. 1f). Suppressed mitochondrial translation is often coupled with a smaller body size[21,22], and knockdown of *C32E8.9* in *mrps-5* RNAi worms eliminated this phenotype as the body size of these worms was similar to those fed with control RNAi (Fig. 1g, h). Taken together, these results highlight that *C32E8.9* is required for the lifespan extension and healthspan benefits observed upon mitochondrial translation inhibition.

### *C32E8.9* is specifically required for mitochondrial translation inhibition-induced longevity without altering the activation of the UPR[mt] gene expressions

Given the crucial role of *C32E8.9* in mediating lifespan extension upon mitochondrial translation inhibition, we wondered whether it plays a

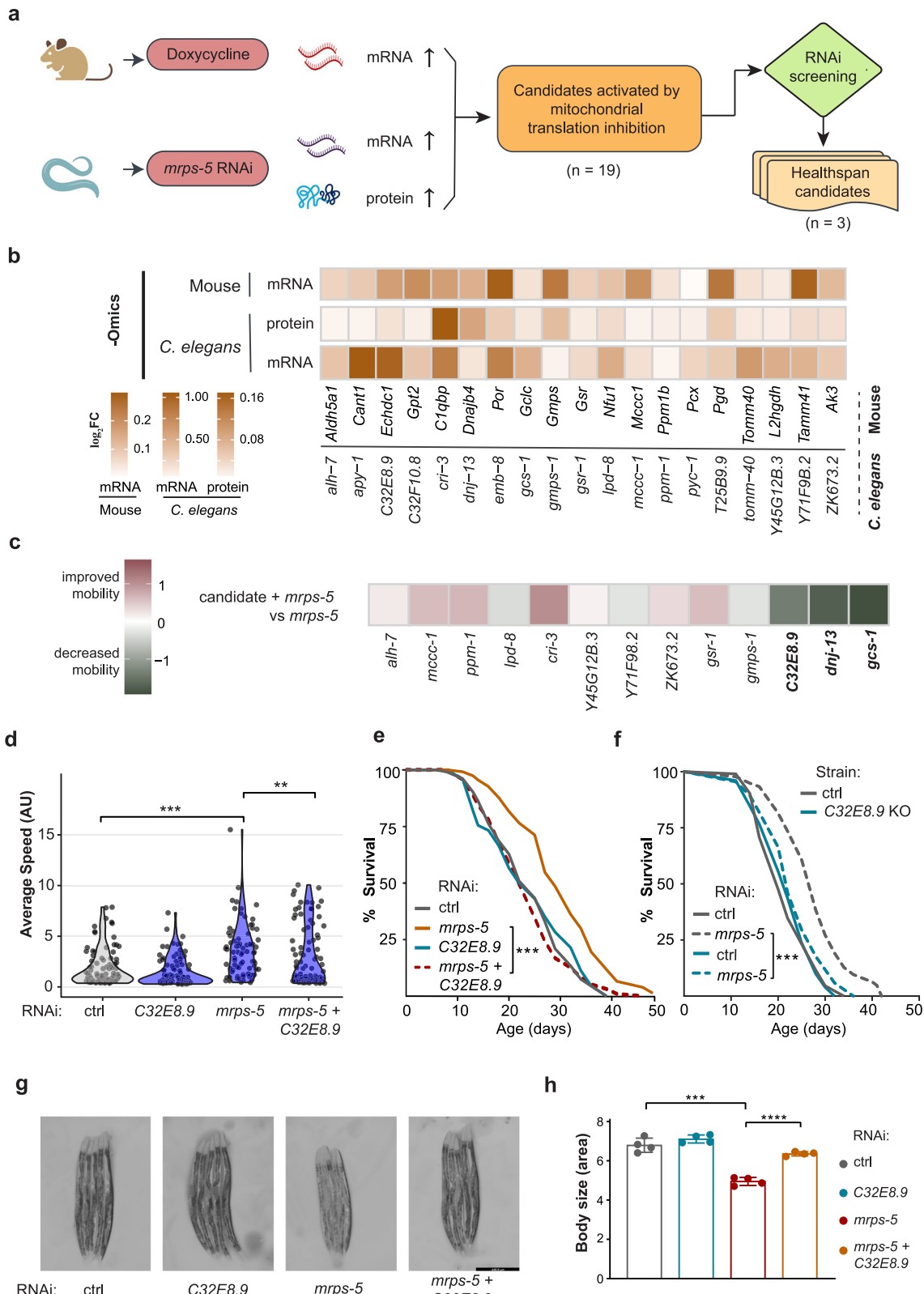

more generalized role in longevity pathways. To address this, we knocked down *C32E8.9* in *cco-1* RNAi worms. Cytochrome c oxidase (or *cco-1*) is a key component of the mitochondrial electron transport chain. Inhibition of *cco-1* is a well-known intervention that, similarly to *mrps-5* RNAi, activates the UPR^mt and extends lifespan in worms[21]. Here, we found that *C32E8.9* was not required for lifespan extension due to *cco-1* deficiency (Fig. 2a). We then tested other established longevity

interventions that do not critically depend on UPR^mt, such as insulin/IGF-1 signaling and caloric restriction[23,24]. Survival analysis clearly indicated that knockdown of *C32E8.9* did not attenuate the lifespan of long-lived *daf-2(e1307)* (insulin/IGF-1) and *eat-2(ad465)* (caloric restriction) mutants (Fig. 2b, c). Based on these results, we conclude that *C32E8.9* is not required for the lifespan extension induced by mitochondrial electron transport chain (*cco-1* RNAi), reduced IIS

**Fig. 1 | *C32E8.9* contributes the beneficial effects induced by mitochondrial translation inhibition. a** Diagram of candidate gene selection and mobility assay-based screening for healthspan candidates. **b** Heatmap of commonly upregulated genes in mice (following doxycycline treatment; at the mRNA level) and *C. elegans* (upon *mrps-5* RNAi; at mRNA and protein level). **c** Heatmap of log$_2$(fold change) of average moving speeds of GMC101 worms in different comparisons at day 4. GMC101 worms were treated with control (HT115), candidate RNAi, *mrps-5* RNAi, or candidate RNAi + *mrps-5* RNAi. Candidates that significantly reduced mobility improved by *mrps-5* RNAi are highlighted in bold. **d** Violin plot representing the average moving speed of GMC101 worms under control (HT115), *C32E8.9* RNAi, *mrps-5* RNAi and *C32E8.9* + *mrps-5* RNAi bacteria at day 4. \*\*\* represents two-sided wilcox.test *p*-value < 0.001. AU arbitrary units. **e** Lifespan curve of N2 worms cultured under control (HT115), *C32E8.9* RNAi, *mrps-5* RNAi and *C32E8.9* + *mrps-5* RNAi bacteria. 323–331 worms were used for each condition in three independent replicate experiments, as specified in Supplementary Data 1. **f** Lifespan analysis of control worms (N2 background strain) and *C32E8.9* knockout (KO) worms under treatment with either control bacteria (HT115) or *mrps-5* RNAi bacteria. 286–296 worms were analyzed per condition across three independent experimental replicates, as detailed in Supplementary Data 1. For (**e**) and (**f**), *p*-values were calculated using the log-rank test to compare each condition with the corresponding control. Statistical significance is indicated as follows: \*\*\* represents *p*-value < 0.001. **g** Representative microscopy image displaying the morphology of N2 worms at day2 under control (HT115), *C32E8.9* RNAi, *mrps-5* RNAi and *C32E8.9* + *mrps-5* RNAi bacteria. RNAi treatment was initiated from the parental stage. Scale bar: 447.2 μm. **h** Quantification of body size of N2 worms at day2 under control (HT115), *C32E8.9* RNAi, *mrps-5* RNAi and *C32E8.9* + *mrps-5* RNAi bacteria. Data from one of three independent experiments is shown. Each point in the boxplot represents a biological replicate. Data are presented as mean values ± SD. Statistical analysis was performed using the Analysis of Variance (ANOVA) test. \*\*\* represents *p*-value < 0.001 and \*\*\*\* represents *p*-value < 0.0001.

(*daf-2(e1307)*), or caloric restriction (*eat-2(ad465)*), but is crucial for lifespan extension induced by *mrps-5* RNAi.

As the UPR$^{mt}$ has been functionally implicated as a mediator of the lifespan extension induced by mitochondrial translation inhibition[6,25], we wondered whether the observed effect of *C32E8.9* inhibition on longevity was governed through attenuation of the UPR$^{mt}$ response. We used *hsp-6p::gfp* reporter worm to examine the UPR$^{mt}$ induction, and we found that inhibition of *C32E8.9* did not affect the fluorescence signal of *hsp-6p::gfp* activated by *mrps-5* RNAi (Fig. 2d, e), suggesting that the lifespan effect of *C32E8.9* RNAi is independent of the UPR$^{mt}$. To further characterize the potential effect of *C32E8.9* RNAi on UPR$^{mt}$ activation, we analyzed the mRNA expression of two key UPR$^{mt}$ genes, *hsp-6* and *clpp-1*, in N2 worms treated with *mrps-5* and/or *C32E8.9* RNAi. Knockdown of *mrps-5* resulted in a non-significant but noticeable trend toward increased mRNA expression of *hsp-6*, and the simultaneous knockdown of *C32E8.9* with *mrps-5* did not affect this trend (Fig. 2f). Additionally, knockdown of *mrps-5* significantly increased the expression of *clpp-1*, and this upregulation remained unchanged with the concurrent knockdown of *C32E8.9* (Fig. 2f). These results are in line with our observations from the *hsp-6p::gfp* reporter worm assays. Taken together, *C32E8.9* is specifically required for mediating mitochondrial translation inhibition coupled with lifespan extension, without significantly altering the activation of the UPR$^{mt}$.

## *C32E8.9* is required for the immune response activated by mitochondrial translation inhibition

To untangle how *C32E8.9* plays a role in mitochondrial translation-mediated lifespan extension without significantly altering UPR$^{mt}$, we performed RNA sequencing on total RNA isolated from day 5 adult worms fed with *mrps-5* RNAi, *C32E8.9* RNAi, or the combination (Fig. 3a). Principal component analysis (PCA) revealed that inhibition of *C32E8.9* alone did not have a pronounced impact on the transcriptome as it overlapped with the controls (Fig. 3b), which is in line with what we observed in terms of its effect on healthspan and lifespan (Fig. 1d, e). Meanwhile, the transcriptome of *mrps-5* RNAi worms and those fed with *C32E8.9* RNAi in addition to *mrps-5* RNAi were distinctly separated from all the other groups. This suggests that *C32E8.9* plays a role in modulating transcriptome alterations induced by *mrps-5* RNAi, as the transcriptome of the double knockdown demonstrates a shift towards the control on the first principal component compared to the effects of *mrps-5* RNAi alone.

We further investigated differentially expressed genes in different groups of comparisons and found that genes are mainly upregulated under *mrps-5* RNAi compared to the control. When comparing the double inhibition of *mrps-5* and *C32E8.9* to the RNAi of *mrps-5* alone, there were fewer upregulated genes and more downregulated genes (Fig. 3c, d). These data indicated that inhibition of *C32E8.9* blunted the transcriptome activated by *mrps-5* RNAi. Subsequently, we performed an over-representation analysis to explore the potential function of genes that were upregulated by *mrps-5* RNAi but reversed by *C32E8.9* knockdown. Here, we found that the top KEGG gene sets enriched in upregulated genes following *mrps-5* RNAi were "MAPK signaling pathway," "Neuroactive ligand–receptor interaction," "Drug metabolism-cytochrome P450," and "Retinol metabolism" (Fig. 3e). These terms were reversed under double knockdown versus *mrps-5* RNAi (Fig. 3f). A similar pattern can be observed in the enrichment of transcripts of cellular components (Supplementary Fig. 2a, b). The immune-related pathways are the top GO biological process gene sets enriched in genes downregulated in double knockdown compared to *mrps-5* RNAi (Fig. 3g, h). This is consistent with the observed reversal of immune-related KEGG pathways. Altogether, these results suggest that *C32E8.9* plays a pivotal role in the activated immune response induced by mitochondrial translation inhibition.

## *sma-4* is required for lifespan extension induced by mitochondrial translation inhibition

Given the observed role of *C32E8.9* in modulating the immune response triggered by mitochondrial translation inhibition, we proceeded to elucidate the immune regulators implicated in this modulation. The MAPK[26,27], JAK/STAT[28], and TGF-β[29] pathways exhibit highly conserved mechanisms in the regulation of immunity, lifespan, and aging from *C. elegans* to mammals[30] (Fig. 4a). Our RNA-seq data demonstrated increased expression of key components in the p38/JNK MAPK pathway (*pmk-1, pmk-3, mek-1, sek-1*) under *mrps-5* RNAi, which were reversed when *mrps-5* is double-knocked down with *C32E8.9*. A similar regulatory pattern was observed in the transcription factors of the JAK/STAT pathway (*sta-1/2*) and co-transcription factors of the TGF-β pathway (*sma-2/3/4*) (Fig. 4b). We hence asked if these activated defense responses were also involved in the lifespan extension upon *mrps-5* RNAi. We measured the lifespans of worm strains in which these defense responses were blocked, together with or without *mrps-5* RNAi. *mrps-5* RNAi still significantly extended lifespan of *pmk-1(km25)*, *mek-1(ks54)/sek-1(qd127)*, and *pmk-3(ok169)* worms (Fig. 4c, d, Supplementary Fig. 2e). Similarly, reduction of *nsy-1* or *sta-2* through RNAi did not shorten the lifespan extended by *mrps-5* RNAi (Fig. 4e, f). Following this, we tested whether *sma-3* and *sma-4*, the two key co-transcription factors of the TGF-β pathway, could affect mitochondrial translation inhibition-induced longevity. While *sma-3* RNAi did not blunt the lifespan extension upon *mrps-5* RNAi (Fig. 4g), knockdown of *sma-4* almost completely blocked the lifespan extension mediated by *mrps-5* RNAi (Fig. 4h). To rule out potential off-target effects of *sma-4* RNAi, we examined the lifespan of *mrps-5* RNAi in the *sma-4* mutant *DR1369* worms. The lifespan analysis revealed that knockdown of *mrps-5* failed to extend the lifespan in *sma-4(DR1369)* worms (Supplementary Fig. 2f). Furthermore, we observed that worms undergoing double knockdown of *sma-4* with *mrps-5* have wider body

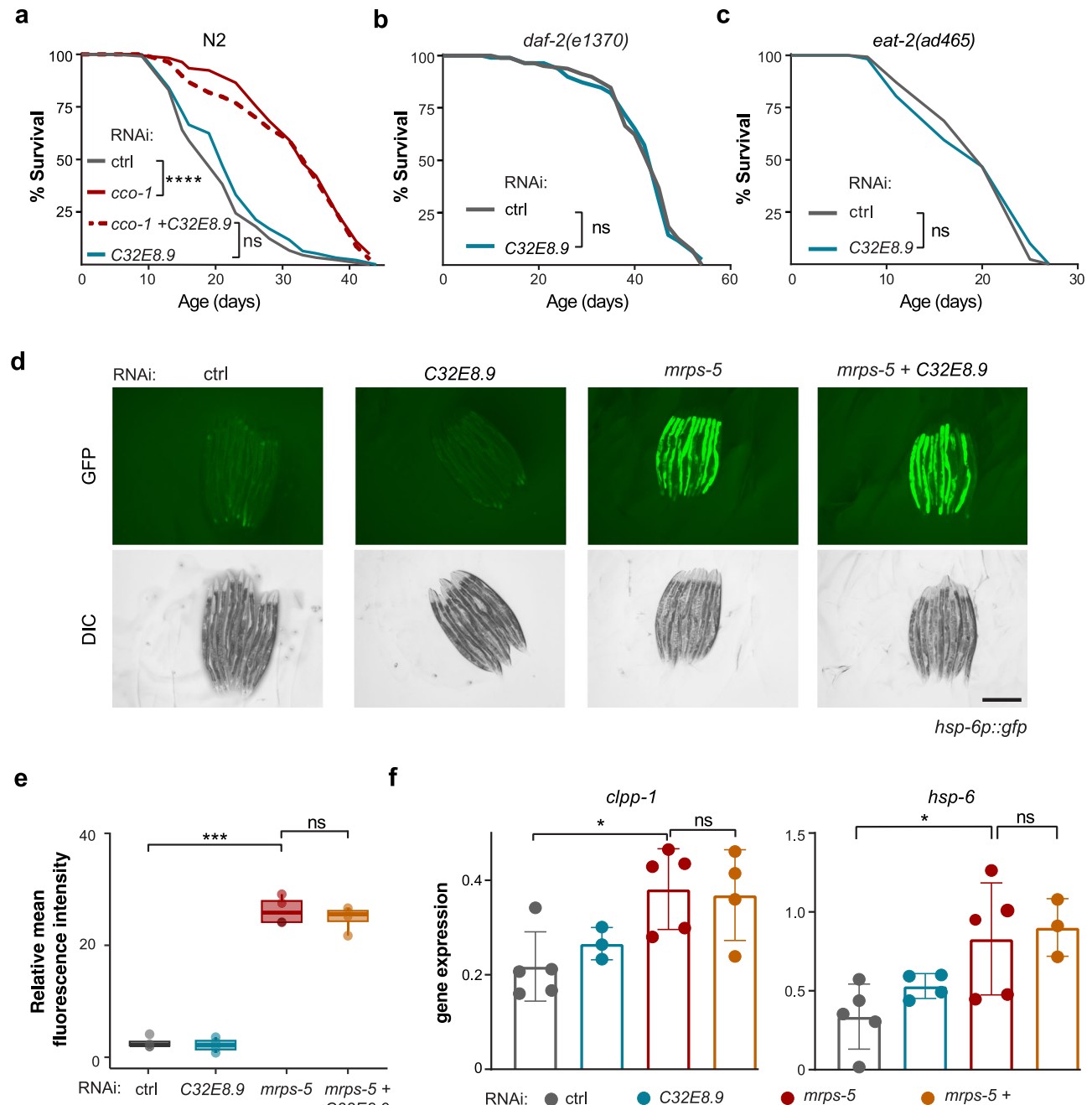

**Fig. 2 | *C32E8.9* is specifically required for *mrps-5* RNAi-induced longevity without altering the activation of the UPR^mt gene expressions. a** Lifespan curves of N2 worms treated with control (HT115), *C32E8.9* RNAi, *cco-1* RNAi, *C32E8.9* RNAi + *cco-1* RNAi. Each condition included 232–246 worms, with data obtained from two independent replicate experiments, as outlined in Supplementary Data 1. **b** Lifespan curves of *daf-2(e1370)* worms treated with control (HT115) or *C32E8.9* RNAi. 219–223 worms were used for each condition in two independent replicate experiments, as specified in Supplementary Data 1. **c** Lifespan curves of *eat-2(ad465)* worms treated with control (HT115) or *C32E8.9* RNAi. Each condition included 235–239 worms, with experiments conducted in two independent biological replicates, as summarized in Data 1. **d** Representative fluorescence images of *hsp-6p::gfp* reporter worms under control (HT115), *C32E8.9* RNAi, *mrps-5* RNAi and

*C32E8.9* + *mrps-5* RNAi bacteria at day 3. Scale bar: 500 μm. **e** Quantification of relative fluorescence intensity of *hsp-6p::gfp* reporter worms under control (HT115), *C32E8.9* RNAi, *mrps-5* RNAi and *C32E8.9* + *mrps-5 RNAi* bacteria at day 3. Each dot represents a biological replicate. Box plots indicate the median (center line), the 25th and 75th percentiles (bounds of the box), and the minimum and maximum values (whiskers). *** represents *p*-value < 0.001. The statistical analysis was performed using the Analysis of Variance (ANOVA) test. **f** Bar plots representing the relative mRNA expression of *clpp-1* (left panel) and *hsp-6* (right panel) in N2 worms under the same conditions as in (**e**). Data from one of three independent experiments is shown, with each dot representing a biological replicate (**e**, **f**). Data are presented as mean values ± SD. ***: *p*-value < 0.001, *: *p*-value < 0.05. The statistical analysis was performed using the Analysis of Variance (ANOVA) test.

width compared to *mrps-5* RNAi alone (Fig. 4i, j). In summary, mitochondrial translation inhibition increased the expression of *sma-4*, which plays a key role in mediating *mrps-5* RNAi-activated traits, not only blocking the lifespan extension but also reversing the slim body size induced by *mrps-5* RNAi.

## *C32E8.9* and *sma-4* mediate the mitochondrial translation inhibition-activated immune response

Given the observed relationship between *C32E8.9*, *sma-4*, and *mrps-5*, where both *C32E8.9* and *sma-4* were required for the lifespan extension by *mrps-5* RNAi, we sought to identify possible interactions between

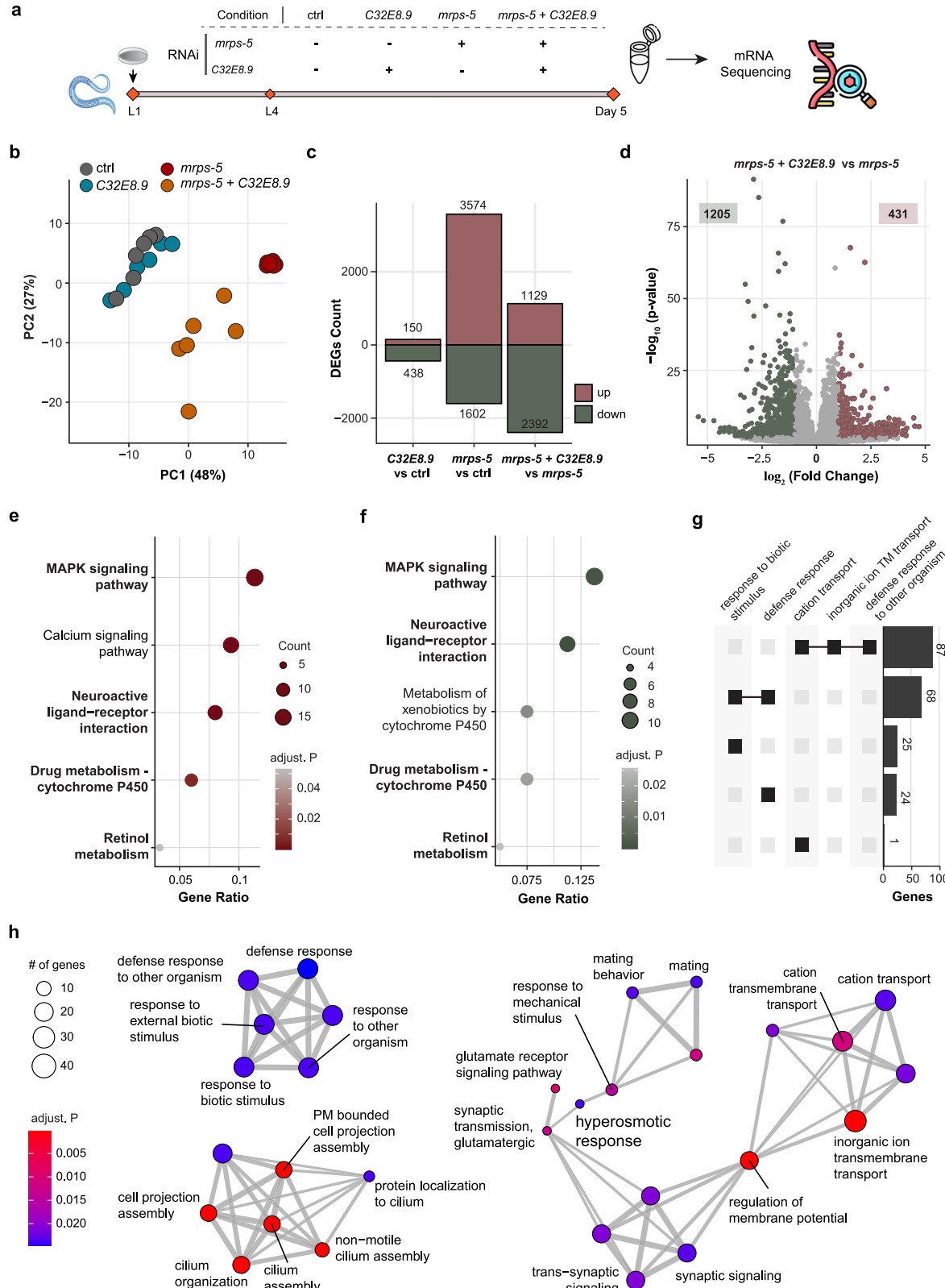

these genes. GeneMANIA is a flexible, user-friendly web interface for generating hypotheses about gene function and analyzing gene lists[31]. It utilizes available public genomics data, proteomics data, and annotation data to predict gene functions and analyze how genes in a gene list are connected to one another with high accuracy[31]. As such, we explored how *C32E8.9*, *sma-4*, and *mrps-5* are connected to one another in GeneMANIA. The network generated from GeneMANIA

showed that no direct associations were reported between *C32E8.9*, *sma-4*, and *mrps-5* so far (Supplementary Fig. 2d), although *mrps-5* co-expressed with the *C32E8.9* and *sma-4* modules through a direct correlation with *ech-2* and *sma-3*, respectively. We next selected genes highly associated with *mrps-5*, *C32E8.9*, and *sma-4* through GeneMA-NIA, and examined their expression profiles in our RNA-seq dataset (Fig. 5a). We found that *sma-4* expression was upregulated upon *mrps-*

**Fig. 3 | *C32E8.9* is required for the immune response activated by mitochondrial translation inhibition. a** Experimental design of mRNA sequencing. Worms fed with RNAi bacteria (control = HT115, *C32E8.9* RNAi, *mrps-5* RNAi, *mrps-5* RNAi + *C32E8.9* RNAi) starting from the L1 phase. **b** PCA plot of the transcriptomic analysis. Samples from *mrps-5* RNAi cluster separately from control samples. Samples from the *mrps-5* + *C32E8.9* double RNAi are projected between the control and *mrps-5* RNAi samples. *C32E8.9* RNAi samples mainly overlap with control samples. **c** Bar plot of the number of significantly differentially expressed genes (adjusted *p*-value < 0.05, absolute log$_2$FC > 0.5) in different comparisons. **d** Volcano plot of genes in *mrps-5* + *C32E8.9* double RNAi vs *mrps-5* RNAi (adjusted *p*-value < 0.05, absolute log$_2$FC > 1). **e** Top 5 over-representation enriched KEGG gene sets in significantly upregulated genes (adjusted *p*-value < 0.05, log$_2$FC > 0.5) in *mrps-5* RNAi

vs control. **f** Top 5 over-representation enriched KEGG gene sets in significantly downregulated genes (adjusted *p*-value < 0.05, log$_2$FC < −0.5) in *mrps-5* + *C32E8.9* double RNAi vs *mrps-5* RNAi. Terms appearing in both (**e**) and (**f**) are highlighted in bold. **g** Upset plot of top 5 enriched biological process gene sets in significantly downregulated genes (adjusted *p*-value < 0.05, log$_2$FC < −0.5) in *mrps-5* + *C32E8.9* double RNAi vs *mrps-5* RNAi. The size of the bar represents the number of genes overlapping among gene sets. **h** Graph representation of the top 30 enriched biological process gene sets of downregulated genes (adjusted *p*-value < 0.05, log$_2$FC < −0.5) in *mrps-5* + *C32E8.9* double RNAi vs *mrps-5* RNAi. Color represents the adjusted *p*-value. The size of the nodes represents the number of genes. The thickness of the edges represents the number of shared genes among the two nodes.

5 RNAi, but this effect was reversed when *C32E8.9* was also knocked down (Supplementary Fig. 2i). To validate this, we performed qPCR, which confirmed the expression pattern observed in RNA-Seq (Supplementary Fig. 1e). Consistent with the expression of *sma-4*, other genes in the TGF-β signaling pathway were similarly upregulated in response to *mrps-5* RNAi and reversed by *C32E8.9* RNAi (Supplementary Fig. 2g, h).

Previous studies discovered that the TGF-β pathway aids in resisting infections by *Pseudomonas aeruginosa* and *Serratia marcescens*[32,33]. The co-transcription factor *sma-4* serves as a core regulator of this pathway. Using a slow-killing assay, we tested whether *sma-4* and/or *C32E8.9* are involved in the innate immune response activated by *mrps-5* RNAi. *mrps-5* RNAi moderately increased the survival rate of worms chronically infected with *Pseudomonas aeruginosa* strain PA01. This protective effect was blunted upon double knockdown of *mrps-5* and *C32E8.9* (Fig. 5b). We also corroborated these results using the *C32E8.9* knockout strain. Consistent with the results obtained using *C32E8.9* RNAi, the enhanced survival conferred by *mrps-5* RNAi in control worms was completely lost in the *C32E8.9* knockout strain (Fig. 5c). A similar pattern was observed in the double knockdown of *mrps-5* and *sma-4*, as seen with the double RNAi of *mrps-5* and *C32E8.9* (Fig. 5d). The innate immune response and the food avoidance response are extensively interconnected, as activated immune responses lead to enhanced food avoidance behavior[34–36]. To further examine the role of *C32E8.9* and *sma-4* in the pathogen response activated by *mrps-5* RNAi, we performed a food avoidance assay, which counts the number of worms residing on and off a bacterial lawn. The food avoidance assay corroborated the results of the slow-killing assay, indicating that inhibition of *C32E8.9* or *sma-4* blocked the food avoidance response induced by *mrps-5* RNAi (Fig. 5e–g). In summary, *C32E8.9* and *sma-4* play a role in mediating the enhanced immune response triggered by *mrps-5* RNAi.

### *C32E8.9* mediates lipidomic changes taking place under mitochondrial translation inhibition

As an ortholog of human ECHDC1, *C32E8.9* is predicted to be involved in fatty acid metabolism (provided by http://ortholist.shaye-lab.org/results)[20]. This association is supported by the enriched gene sets derived from genes interacting with *C32E8.9* in *C. elegans*, as annotated in GeneMANIA, primarily relating to catabolic processes of lipids and fatty acids (Fig. 6a, Supplementary Fig. 3a). Therefore, we used our established liquid chromatography–mass spectrometry (LC–MS)-based semi-targeted lipidomics platform to uncover how *C32E8.9* regulates the lipidome in the context of *mrps-5* RNAi (Supplementary Fig. 3b). We were able to annotate around 2000 unique complex lipid species belonging to a wide range of lipid classes. PCA analysis revealed partial overlap of *mrps-5* RNAi and control suggesting the tested lipidome maintains a similar profile under mitochondrial translation inhibition (Fig. 6b). Meanwhile, samples subjected to double RNAi targeting *mrps-5* and *C32E8.9* were projected between those treated with single *C32E8.9* RNAi and *mrps-5* RNAi, indicating a

partial reversal of the lipid program maintained by *mrps-5* RNAi upon double RNAi treatment (Fig. 6b).

Our lipid class enrichment analysis revealed significant alterations following *mrps-5* RNAi, particularly in triglycerides (TGs), which constituted the largest subset of enriched lipids (Fig. 6c). Further over-representation enrichment analysis indicated that the lipids reduced in *mrps-5*/*C32E8.9* double RNAi compared to *mrps-5* RNAi alone were significantly enriched in TGs and 2-acyl lysophosphatidylcholine (LPC) (Fig. 6d). Notably, these two lipid classes were also enriched in the *mrps-5* RNAi condition compared to controls (Fig. 6c). To better understand the specific changes in each condition, we conducted a differential abundance analysis of lipids (Fig. 6e, Supplementary Fig. 3d). Among the conditions tested, the single knockdown of *C32E8.9* had the most substantial impact, resulting in 189 lipids accumulating and 352 being depleted. Details of these lipid changes are presented in Supplementary Fig. 3f–h. Taking a closer look at the TGs, an increase of longer polyunsaturated TGs coupled with a decrease in shorter, less saturated TGs was observed when comparing double RNAi of *mrps-5* and *C32E8.9* to *mrps-5* RNAi (Fig. 6f, Supplementary Fig. 3e). We also observed a similar changed pattern in saturation and length in other lipids, including DGs (diacylglycerols) and PCs (phosphatidylcholines) (Fig. 6g, h), though the changes in DGs and PCs were not as obvious as in TGs. This indicated that the knockdown of *C32E8.9* in the context of mitochondrial translation inhibition led to a shift in lipidome from homeostasis to a program with decreased short saturated and increased longer unsaturated fatty acids. Taken together, the observed lipidome changes suggest that *C32E8.9* plays a crucial role in regulating fatty acid elongation and saturation, and mediating lipidomic changes taking place under mitochondrial translation inhibition.

### Overexpression of *C32E8.9* partially replicates the beneficial effects of mitochondrial translation inhibition

Our data so far demonstrate that *C32E8.9* is essential for mediating the beneficial effects observed with *mrps-5* RNAi. Notably, inhibition of mitochondrial translation resulted in a significant increase in *C32E8.9* expression. To further investigate whether overexpression of *C32E8.9* could replicate these benefits, we generated a *C32E8.9* overexpression (OE) strain tagged with green fluorescent protein (GFP). Representative fluorescence images of control worms (with the same genetic background) and the *C32E8.9* OE strain are provided in Supplementary Fig. 3i. In the *C32E8.9* OE strain, GFP was detected throughout the body, while mCherry expression was restricted to the pharynx, confirming successful transgene expression. We observed strong GFP fluorescence in the pharynx, which we attribute to spectral bleed-through between the GFP and mCherry channels (Supplementary Fig. 4). We first evaluated the lifespan of the *C32E8.9* OE strain compared to controls with the same genetic background. The *C32E8.9* OE strain exhibited a significant lifespan extension, consistent with the effects observed upon *mrps-5* RNAi treatment. However, the extent of lifespan extension in *C32E8.9* OE strain was not as pronounced as with *mrps-5* RNAi alone (Fig. 7a).

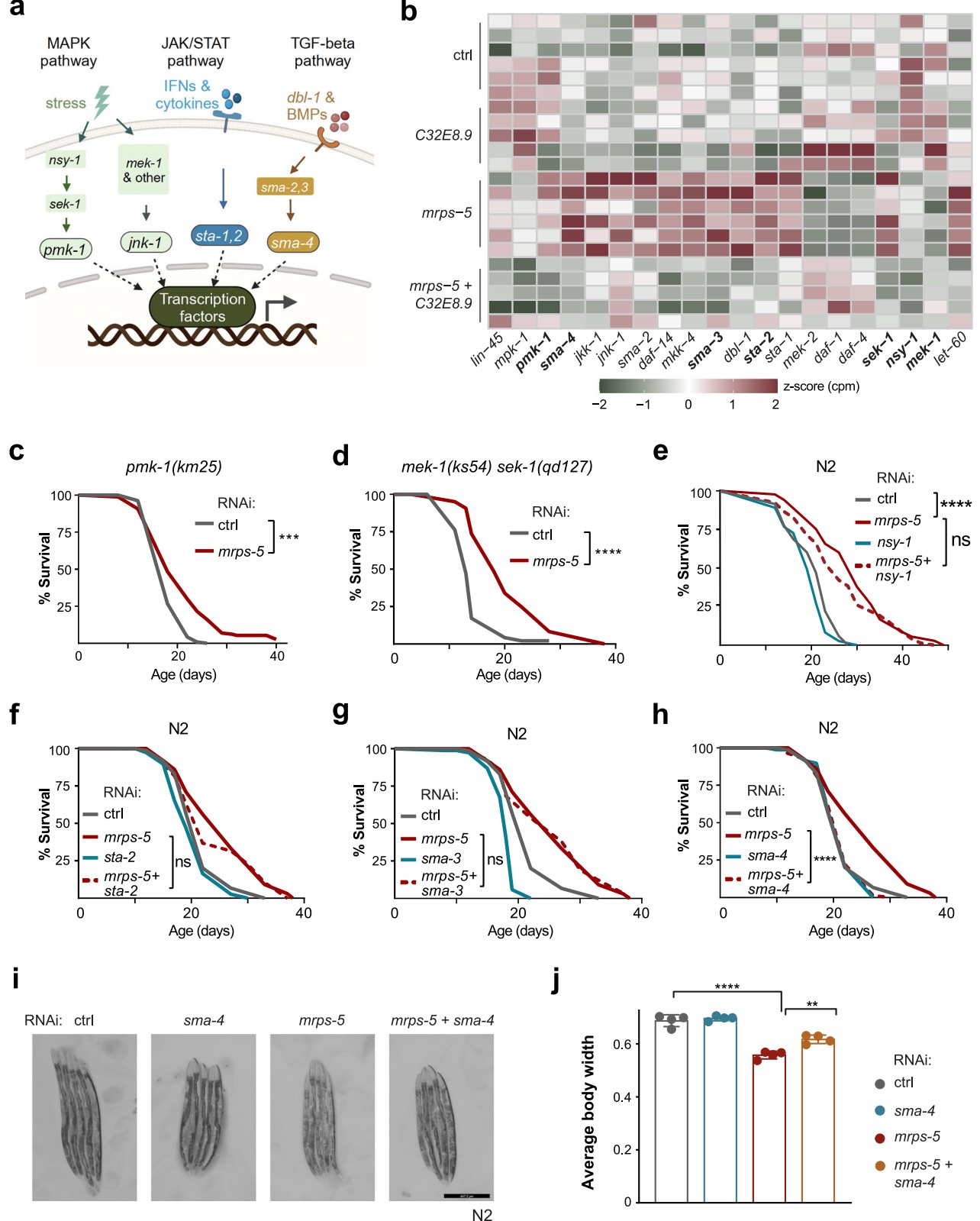

Beyond lifespan, we also evaluated whether the overexpression of *C32E8.9* enhances immune function by performing a PA01 resistance assay. Consistent with the lifespan results, we observed that *C32E8.9* OE worms displayed increased survival compared to controls (Fig. 7b). Overall, the data obtained from the *C32E8.9* overexpression experiments indicate overexpressing *C32E8.9* could partially replicate the beneficial effect of *mrps-5* RNAi, supporting the functional

role of this gene in mediating the beneficial effects initially observed with *mrps-5* RNAi.

Given the observations that knockdown of *C32E8.9* blocked the longevity effect (Fig. 1e) and blunted the enhanced immune response (Figs. 3h, 4b, 5c, d) induced by *mrps-5* RNAi without altering the UPR$^{mt}$ (Fig. 2), and that overexpression of *C32E8.9* could partially recapitulate the beneficial effects of *mrps-5* RNAi (Fig. 7), our data support that

**Fig. 4 | *sma-4* is required for lifespan extension upon mitochondrial translation inhibition. a** Schematic of conserved immune regulators between mammals and *C. elegans*. Created in BioRender. Hu, M. (2025) https:// BioRender.com/2jezyyf. **b** Heatmap of mRNA expression of immune regulators (as shown in (**a**)) in four RNAi conditions, i.e., *nsy-1, sek-1, pmk-1, mek-1, sta-1, sta-2, dbl-1, sma-2, sma-3, sma-4*. Fill color represents the Z-score transformed counts per million (CPM) gene expression for each gene across all samples. For visualization purposes, values exceeding 2 were capped at 2, while values below −2 were capped at −2. **c** Lifespan measurements in *pmk-1(km25)* mutants, worms were fed with control (solid gray line) or *mrps-5* RNAi (solid red line). 240–300 worms were used for each condition in two independent replicate experiments, as specified in Supplementary Data 1. **d** Lifespan measurements in *mek-1(ks54)/sek-1(qd127)* mutants, same conditions as in (**c**). 203–221 worms were used for each condition in two independent replicate experiments, as specified in Supplementary Data 1. **e** Lifespan measurements of N2 worms fed with RNAi bacteria. Control (solid gray line), *mrps-5* RNAi (solid red line), *nsy-1* RNAi (cyan), and *nsy-1* with *mrps-5* double RNAi (dashed red line). 191–198 worms were used for each condition in two independent replicate experiments, as specified in Supplementary Data 1. **f** Lifespan measurements of N2 worms fed with RNAi bacteria. *sta-2* RNAi (cyan), and *sta-2* with *mrps-5* double RNAi (dash red line).

185–192 worms were used for each condition in two independent replicate experiments, as specified in Supplementary Data 1. **g** Lifespan measurements of N2 worms fed with RNAi bacteria. *sma-3* RNAi (cyan), and *sma-3* with *mrps-5* double RNAi (dash red line). 163–175 worms were used for each condition in two independent replicate experiments, as specified in Supplementary Data 1. **h** Lifespan measurements of N2 worms fed with RNAi bacteria. *sma-4* RNAi (cyan), and *sma-4* with *mrps-5* double RNAi (dash red line). 170–173 worms were used for each condition in two independent replicate experiments, as specified in Supplementary Data 1. **c**–**h** *P*-values represent a comparison with the controls calculated using the log-rank test. *** represents *p*-value < 0.001, **** represents *p*-value < 0.0001, "ns" represents not significant. **i** Representative microscopy images displaying the morphology of N2 worms on day 2 of adulthood. RNAi treatment was initiated from the parental stage. Scale bar: 447.2 μm. **j** Quantification and statistical analysis of average body width of N2 worms under the same conditions as in (**i**). Data from one of three independent experiments is shown, with each dot representing a biological replicate. **** represents *p*-value < 0.0001, ** represents *p*-value < 0.01. Data are presented as mean values ± SD. The statistical analysis was performed using the Analysis of Variance (ANOVA) test.

---

*C32E8.9* mediates the lifespan extension of *mrps-5* RNAi through regulating the immune response, possibly by reprogramming the lipidome, especially TG's saturation and length (Fig. 7c).

## Discussion

Through comparative analysis of mouse transcriptome and worm proteome-transcriptome following doxycycline treatment and *mrps-5* RNAi, respectively, we identified a panel of candidates potentially responsible for the beneficial effects observed upon mitochondrial translation inhibition. We then ascertained that *C32E8.9*, a proposed lipid metabolic enzyme that we discovered in this panel, is required for lifespan extension via mitochondrial translation suppression. We showed that inhibition of *C32E8.9* completely abolished the lifespan extension and improved the healthspan conferred by *mrps-5* RNAi, importantly without altering UPR^mt induction. This response was specific to mitochondrial translation inhibition and did not manifest in other longevity interventions. Mitochondrial translation suppression enhanced immune response against pathogen infection, and we show that *C32E8.9* is responsible for this immune regulation by mediating *sma-4*, which is essential for supporting lifespan extension upon mitochondrial translation inhibition. Furthermore, we showed the crucial role of *C32E8.9* in maintaining lipid balance, especially TGs' saturation and length, which is closely correlated to immune response and lifespan extension under mitochondrial translational suppression.

The activation of UPR^mt has been widely recognized as a common feature linking longevity pathways in long-lived mutants of *C. elegans*, particularly those stemming from mitochondrial dysfunction[6,8,37]. *atfs-1* senses mitochondrial stress and communicates with the nucleus during UPR^mt, whereby nuclear encoded mitochondrial chaperones such as *hsp-6* and *hsp-60* are activated by *atfs-1* in response to misfolded proteins within the mitochondria[38]. Yet, a consensus regarding the model in which UPR^mt plays a causal role is still elusive. For example, deletion of *atfs-1*, which is required for induction of the UPR^mt, fails to impede lifespan extension caused by certain specific UPR^mt inducer genes[15]. Furthermore, constitutive activation of the UPR^mt by gain of function mutations in *atfs-1* can be uncoupled from lifespan extension in *C. elegans*[15,39]. These conflicting results suggest the possibility of concurrent mechanisms in addition to UPR^mt contributing to mitochondrial longevity interventions[39]. One of the key findings of our work is that *C32E8.9* abolishes mitochondrial translation inhibition extended lifespan without altering UPR^mt activation. Mitochondrial translation suppression elevated the expression of *C32E8.9*, and *C32E8.9* RNAi blocked its beneficial effects. However, knockdown of *C32E8.9* did not change either the protein (measured by fluorescence) or mRNA levels of *hsp-6* (measured by qPCR), a well-

established indicator of UPR^mt activation[8]. Furthermore, RNA-seq data revealed that the expression pattern of *hsp-6* and other *atfs-1* target genes (*F15B9.1O, cyp-14A4*, and *C07G1.7*) are consistent with the qPCR results (Supplementary Fig. 2c). As such, our work suggests a novel mechanism, regulated by *C32E8.9*, is required for the beneficial longevity effects of mitochondrial translation inhibition, without altering UPR^mt.

Previous work has shown that knockdown of the mitochondrial electron transport chain (ETC) subunit *cco-1* in neurons can induce the UPR^mt in distal cell types. This results in beneficial effects for the whole organism, including increased lifespan[8]. Here, we find that inhibition of *C32E8.9* did not block the lifespan of the long-lived *cco-1* deficiency mutant. Inhibition of *C32E8.9* specifically counteracted the lifespan extension achieved by mitochondrial translation suppression. This emphasizes the regulatory mechanism of *C32E8.9*, particularly in the context of mitochondrial translation. This discovery supports the idea that mechanisms governing the extension of lifespan may vary significantly depending on the type of mitochondrial stress[15,40].

During aging, *C. elegans* exhibits physical deterioration and an increased proliferation of bacteria within the intestine[41]. This is coupled with an increased vulnerability to bacterial infections, generally referred to as innate immunosenescence, an important contributing factor to the death of nematodes at older ages[42]. Although *C. elegans* lacks an adaptive immune system—crucial for mammalian immunity—it has an innate immune system regulated by evolutionarily conserved signaling pathways, including MAPK, JAK/STAT, and TGF-β pathways[30,43]. Previous analysis of transcriptional changes in long-lived mutants revealed a strikingly significant overlap between genes associated with resistance to bacterial pathogens and those strongly correlated with extended longevity[44]. Recent studies have indicated that p38 MAPK-dependent regulation is critical to promote longevity via supporting neuronal integrity, epidermal lysosomal structures and regulating innate immune responses[27,45]. Similarly, our RNAseq data revealed a broad spectrum of immune genes activated when mitochondrial translation is repressed, primarily in the p38 MAPK, JAK-STAT, and TGF-β pathways. Enrichment analysis also revealed that the MAPK pathway is among the top activated terms in the context of mitochondrial translation inhibition. Moreover, its activation is reversed by inhibition of *C32E8.9*. Loss of PMK-1 (transcription factor of p38 MAPK) compromises the extended lifespan of *daf-2(e1370)* (insulin/insulin-like growth factor (IGF) receptor longevity mutant)[45]. Unexpectedly, we found that *mrps-5* RNAi can still prolong the lifespan of worms with *pmk-1* or *mek-1/sek-1* deficiencies. This observation aligns with our hypothesis that the mechanisms governing lifespan

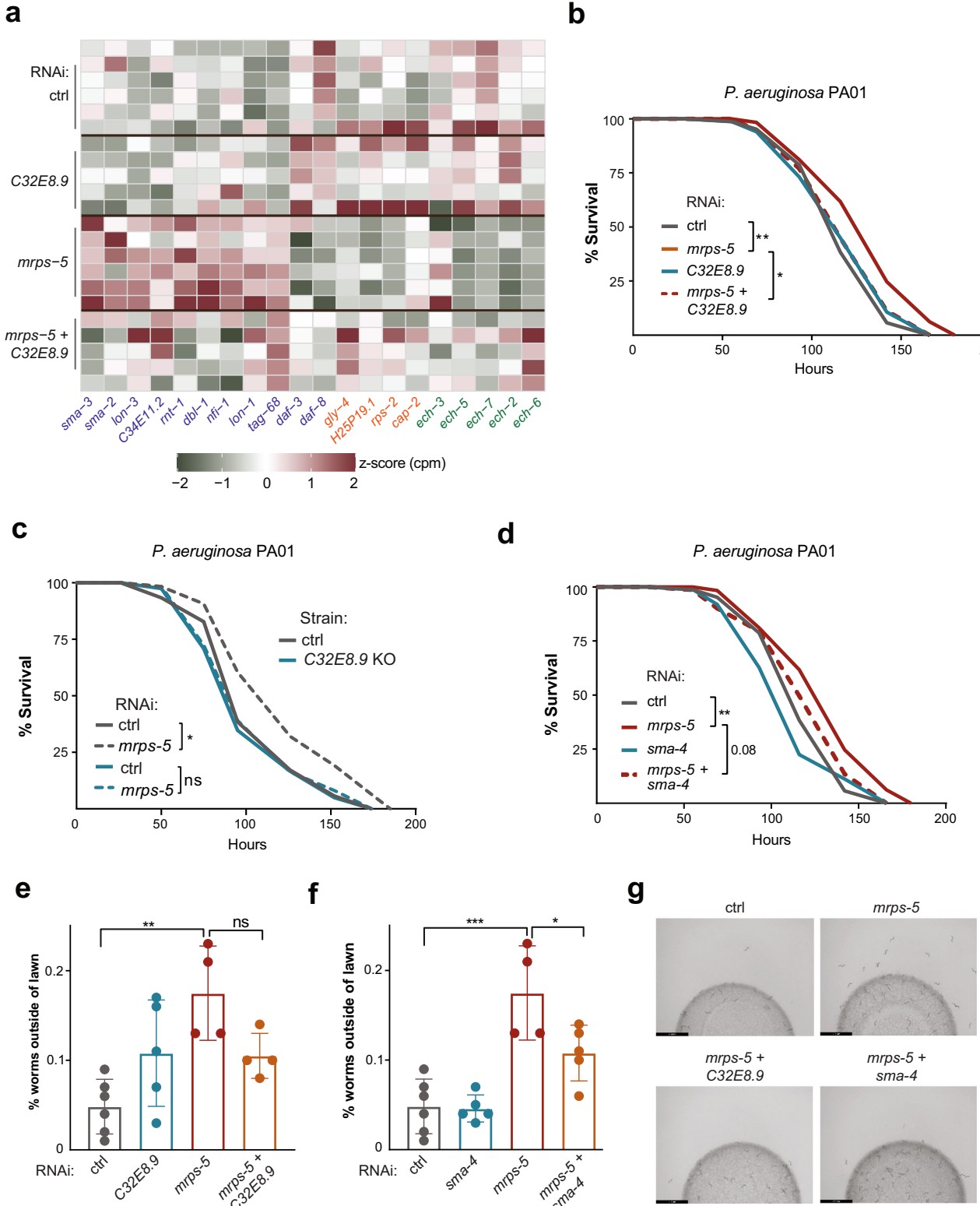

extension vary considerably depending on the type of longevity interventions.

Through inhibition of critical regulators within conserved pathways (p38 MAPK, JAK-STAT, and TGF-β) governing innate immunity, we uncovered that *sma-4*, acting as the co-transcription factor in the TGF-β pathway, is required for the lifespan extension and immune resistance resulting from *mrps-5* RNAi. Our qPCR and RNA-seq data demonstrated that *C32E8.9* RNAi reverses the upregulated mRNA

expression of *sma-4* induced by *mrps-5* RNAi. While this suggests a regulatory connection, the regulation of expression does not necessarily indicate a functional interaction. It remains unclear whether *sma-4* functions downstream of or in parallel with *C32E8.9* in contributing to the beneficial effects of mitochondrial translation inhibition. To address this question, additional functional assays, such as testing whether suppressing *sma-4* blocks the functions of *C32E8*.9, will be required. Future studies will focus on a more detailed characterization

**Fig. 5 | *C32E8.9* and *sma-4* are required for immune response activated by mitochondrial translation inhibition. a** Heatmap showing mRNA expression of genes associated with between *sma-4*, *C32E8.9* or *mrps-5* (as depicted in Fig. S2d). Text color represents clusters from (**a**): purple for *sma-4* cluster, red for *mrps-5* cluster and green for *C32E8.9* cluster. **b** Survival curves of *C. elegans* in a slow-killing model (chronic *P. aeruginosa* PA01 infection). N2 worms treated with control (HT115), *C32E8.9* RNAi, *mrps-5* RNAi, or *C32E8.9* RNAi + *mrps-5* RNAi. For each condition, 193–195 worms were used, with results derived from two independent replicate experiments, as described in Supplementary Table S1. **c** Survival curves of slow-killing assay (chronic *P. aeruginosa* PA01 infection) of control worms (N2 background strain) and *C32E8.9* knockout (KO) worms under treatment with either control bacteria (HT115) or *mrps-5* RNAi bacteria. 169–171 worms were analyzed per condition, with data collected from two independent replicate experiments, as specified in Supplementary Table S1. **d** Survival curves of *C. elegans* in a slow-killing model (chronic *P. aeruginosa* PA01 infection). N2 worms treated with control

(HT115), *sma-4* RNAi, *mrps-5* RNAi, or *sma-4* RNAi + *mrps-5* RNAi. 193–198 worms were analyzed for each condition across two independent replicate experiments, as specified in Supplementary Table S1. **b**–**d** A representative result of 2 independent experiments is shown. log-rank test method was used for survival analyses, * represents *p*-value < 0.05, ** represents *p*-value < 0.01. **e**, **f** Bar plot of food avoidance assay. *Y*-axis represents the percentage of worms outside of the circular OP50 bacterial lawn in the indicated RNAi condition (control, *C32E8.9* RNAi, *mrps-5* RNAi, double RNAi *mrps-5* + *C32E8.9*, *sma-4* RNAi, double RNAi *mrps-5* + *sma-4*). Data from one of three independent experiments is shown, with each dot representing a biological replicate. Data are presented as mean values ± SD. The statistical analysis was performed using an Analysis of Variance (ANOVA) test, *** represents *p*-value < 0.001, ** represents *p*-value < 0.01, * represents *p*-value < 0.05, "ns" represents not significant. **g** Representative plots of food avoidance assay in (**e**, **f**). Scale bar: 2.9 mm. Plots showed half of the plates scored.

of how immune responses, *sma-4* and *C32E8.9* interact and how they contribute to the observed beneficial effects of mitochondrial translation inhibition.

Mitochondrial translation inhibition led to elevated expression levels of genes within the TGF-β pathway, and this effect was reversed by *C32E8.9* RNAi. This further underscores the significance of *C32E8.9* in controlling the favorable outcomes of mitochondrial translation inhibition. The important role of TGF-β signaling in controlling body size and male tail development in *C. elegans* has been well documented[46]. This pathway also plays a central role in regulating immunity through the control of gut microbial proliferation and regulation of antimicrobial peptide expression[29,32]. We further revealed a pivotal role of immune response regulated by TGF-β pathway in contributing to lifespan extension following *mrps-5* RNAi, providing additional understanding of the mechanism and underscoring the beneficial effects of mitochondrial translation repression.

The closest ortholog of nematode *C32E8.9* in human is *ECHDC1*, coding for ethylmalonyl-CoA decarboxylase 1. ECHDC1, as a key modulator of fatty acid metabolism, performs a metabolite repair function to prevent the formation of branched fatty acids, especially ethyl-branched lipids[20,47]. Here, we found that repression of *C32E8.9* leads to profound lipidome reprogramming shifting from short saturated to long, more unsaturated lipids, especially manifesting in TGs. These results demonstrate the crucial role of *C32E8.9* in lipid homeostasis. Lipidomics analysis revealed that lipid saturation and length remain stable following *mrps-5* RNAi, concurrent with the extension of lifespan. Nevertheless, a distinct pattern is observed, indicating that both the saturation and length of TGs shifted toward longer and higher levels of unsaturation when mitochondrial translation was doubly repressed alongside *C32E8.9*. This pattern was also observed in DGs and PCs. Given that the lifespan extension following *mrps-5* RNAi is blocked when double inhibition *C32E8.9* and *mrps-5*, we suggest that the regulation of lipid saturation homeostasis by *C32E8.9* plays a crucial role in sustaining the lifespan extension resulting from mitochondrial translation suppression. Previous studies have reported that alterations of single lipid species, such as ceramide and cholesterol, can influence the immune response in *C. elegans*[48,49]. The change of lipid saturation not only impacts membrane fluidity, flexibility, selective permeability, and peroxidation, but also plays a crucial role in the mechanism of signal transduction[50]. It is hence not surprising that the alteration in lipid saturation levels can impact the lifespan and immune response enhanced by mitochondrial translation suppression. Given the complexity of lipid metabolism, where lipids exist in diverse configurations with varying chain lengths, degrees of saturation, and double bond positions—and considering that their interconversion involves numerous, often promiscuous, enzymes—it is challenging to definitively attribute the beneficial effects of mitochondrial translation suppression to specific lipid changes driven by *C32E8.9*. This complexity complicates our understanding of how these modifications

directly contribute to lifespan extension. While our findings suggest that inhibiting mitochondrial translation enhances longevity, pinpointing the exact lipid alterations responsible for these effects remains elusive. To address these gaps, further investigation is needed into how specific lipid modifications influence immune responses and aging. Future studies will focus on a more detailed characterization of lipid profiles and their downstream effects on immune signaling, stress responses, and metabolic pathways regulated by *C32E8.9*. By exploring these connections, we aim to elucidate the precise mechanisms through which mitochondrial function impacts lipid metabolism and contributes to the observed longevity benefits.

While our findings strongly support the role of *C32E8.9* in mediating lifespan extension through immune and lipid regulation, we acknowledge certain limitations. RNAi efficiency in the double knockdown condition, particularly for *C32E8.9* RNAi, was slightly reduced compared to the single knockdown (Supplementary Fig. S1c). While this reduction was not statistically significant, it may still have biological relevance. Although the inclusion of mutants in our study alleviates concerns about RNAi efficiency, we recognize this as a potential caveat in fully interpreting the role of *C32E8.9*. Future studies utilizing conditional knockout models and endogenous tagging approaches will be valuable in further validating these findings.

In summary, our results provide clear evidence for the role of *C32E8.9* in lifespan extension resulting from impaired mitochondrial translation by modulating innate immunity and lipid reprogramming. *C32E8.9* is activated following mitochondrial translation inhibition; it interconnects lipid saturation balance with immune response and lifespan extension upon mitochondrial translation inhibition.

## Methods

### *C. elegans* strains and maintenance
*C. elegans* strains used in this study are as follows: Bristol N2, CB1370 [*daf-2(e1370)*], DA465 [*eat-2(ad465)*], KU25 [*pmk-1(km25)*], BS3383 [*pmk-3(ok169)*], FK171 [*mek-1(ks54)/sek-1(qd127)*], SJ4100 [*zcIs13[hsp-6p::gfp + lin-15(+)]*], GMC101 [*dvIs100 (unc-54p::A-beta-1-42::unc-54 3′-UTR + mtl-2p::GFP)*]. These strains were obtained from the *Caenorhabditis* Genetic Center (CGC). All the *C. elegans* strains were maintained at 20 °C on standard nematode growth medium (NGM) plates seeded with *E. coli* OP50 unless otherwise indicated. *E. coli* OP50 is cultured overnight in Luria Broth (LB) medium at 37 °C.

### RNAi treatment
All RNAi clones were obtained from the Ahringer RNAi library[51] and confirmed by sequencing. Unless specifically stated, RNAi bacterial feeding experiments were performed from L1 as described[6]. Briefly, gravid adult worms were synchronized by hypochlorite treatment, then plated on NGMi plates (NGM plates with 2 mM IPTG and 25 mg/mL carbenicillin) with a bacterial lawn of either *E. coli* HT115 (RNAi control strain, containing an empty vector) or *mrps-5*, *cco-1*, *C32E8.9*,

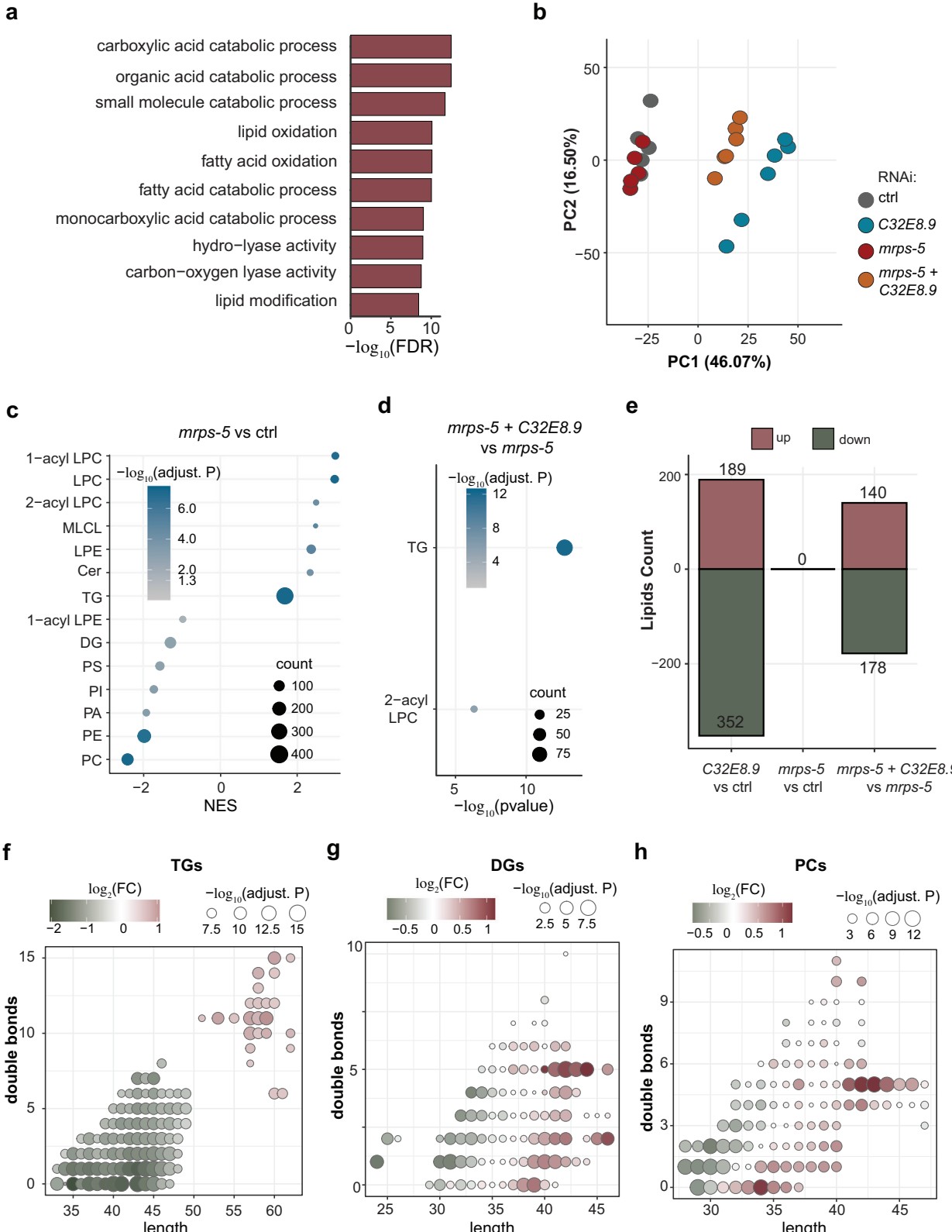

**Fig. 6 | *C32E8.9* mediates lipidomic changes taking place under mitochondrial translation inhibition. a** Bar plot of enriched gene sets in genes interacting with *C32E8.9* in *C. elegans* annotated in GeneMANIA. **b** PCA plot of the lipidome. **c** Lipid classes enrichment analysis of lipids significantly upregulated in *mrps-5* RNAi vs control (adjusted *p*-value < 0.05, absolute log2FC > 1), NES represents 'Normalized Enrichment Score.' **d** Lipid class over-representation analysis of lipids significantly downregulated (adjusted *p*-value < 0.05, absolute log2FC < −1) in *mrps-5* + *C32E8.9* double RNAi vs *mrps-5* RNAi. **e** Bar plot of the number of significantly altered lipids (adjusted *p*-value < 0.05, absolute log2FC > 1) in different comparisons. **f** Dot plot of significantly changed TGs in *mrps-5* + *C32E8.9* double RNAi vs *mrps-5* RNAi comparison. **g** Dot plot of significantly changed DGs in *mrps-5* + *C32E8.9* double RNAi vs *mrps-5* RNAi comparison. **h** Dot plot of significantly changed PCs in *mrps-5* + *C32E8.9* double RNAi vs *mrps-5* RNAi comparison. For (**f**–**h**), *X*-axis represents the length of the individual TG species. *Y*-axis represents the number of double bonds in each of the TG species.

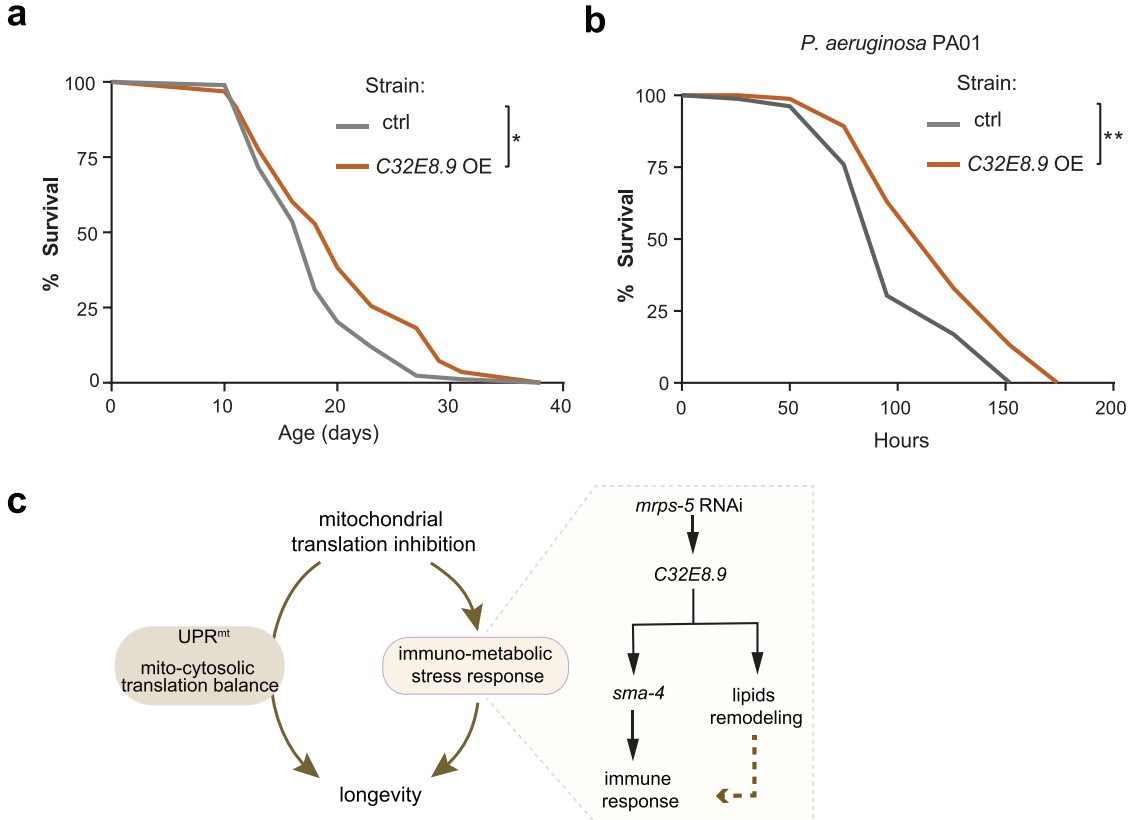

**Fig. 7 | Overexpressing *C32E8.9* partially mimics the beneficial effect of mito-chondrial translation inhibition. a** Lifespan curves of control worms (N2 background strain) and *C32E8.9* expression strain (OE) cultured with OP50. 195 worms were analyzed per condition, with data collected from two independent replicate experiments, as specified in Supplementary Data 1. **b** Survival curves of slow-killing assay (chronic *P. aeruginosa* PA01 infection) of control worms (N2 background strain) and *C32E8.9* expression strain (OE). 168–170 worms were analyzed per condition across two independent experimental replicates, as detailed in Supplementary Table S1. For (**a**) and (**b**), *p*-values were calculated using the log-rank test to compare each condition with the corresponding control. Statistical significance is indicated as follows: ** represents *p*-value < 0.01, * represents *p*-value < 0.05. **c** Hypothetical scheme depicting how immuno-metabolic stress response orchestrated by *C32E8.9* contributes to longevity from mitochondrial translation.

*nsy-1, sta-2, sma-3, sma-4, zip-2*, candidates genes (*ZK673.2, alh-7, cri-3, dnj-13, gcs-1, gmps-1, gsr-1, Y45G12B.3, mccc-1, lpd-8, ppm-1, Y71F9B.2*) RNAi bacteria. For double RNAi experiments, two bacterial cultures were mixed to a specified ratio normalized to optical densities at 600 nm.

### CRISPR/Cas9-mediated knockout of *C32E8.9*
To generate a knockout of the *C32E8.9* gene in *C. elegans*, we employed CRISPR/Cas9-mediated gene editing. The specific target was the *C32E8.9.1* transcript, with the goal of introducing a frameshift and a premature stop codon to disrupt its function. The wild-type background strain (N2) was used for the knockout experiments. The CRISPR/Cas9 system was designed and executed in collaboration with Sunnybiotech, which carried out the sgRNA design, plasmid construction, microinjection, and screening for precise gene deletion. To verify the knockout, quantitative PCR was performed on the edited strains (see Supplementary Fig. 1f). The results confirmed a complete loss of *C32E8.9* mRNA expression, indicating a successful knockout.

### Construction of the stable transgenic *C32E8.9* OE strain
To generate a stable overexpression strain of the *C32E8.9* gene in *C. elegans*, we employed a plasmid-based microinjection approach, followed by genomic integration via X-ray irradiation. The overexpression construct included: Transgene: Pdpy-30-*C32E8.9* cDNA-3xGGSG-GFP-unc-54 3'UTR. Reporter Marker: Pmyo-2-mCherry for pharyngeal expression, used to visually confirm transgene presence. Since little is currently known about the tissue-specific expression of *C32E8.9*, we utilized *dpy-30*, a ubiquitously expressed promoter, to examine potential expression patterns without introducing bias. Microinjections were performed using the transgene plasmid at a concentration of 20 ng/µL, co-injected with the Pmyo-2::mCherry marker into wild-type N2 worms. Following microinjection, worms carrying extrachromosomal arrays were exposed to X-ray irradiation to integrate the transgene into the genome. Stable integrated lines were isolated by screening for consistent mCherry expression in the pharynx, indicating successful integration. Sunnybiotech provided comprehensive support, including plasmid construction, microinjections, X-ray irradiation of the extrachromosomal lines, and the isolation of stable integrated strains. The integration and overexpression were confirmed using GFP fluorescence and mCherry expression (RFP) in the pharynx (Supplementary Fig. 3i) and representative zoom-in plots in Supplementary Fig. 4.

### Mobility assay and RNAi Screen
GMC101 worms were used to measure healthspan by tracking animals' mobility. Gravid adult GMC101 worms were synchronized using alkaline hypochlorite treatment and incubated in M9 buffer overnight to get L1 worms. Worms were fed with described RNAi bacteria. Animals were maintained at 20 °C from L1 to L4 stage. From the L4 stage, worms were transferred to NGMi plates with 5-Fluorouracil (5-FU, Sigma) and cultured at 25 °C. The L4 stage was counted as day 0 of life. To track the moving speed of GMC101, ~50 day 4 adult worms were transferred to 3 cm NGMi plates without bacteria. Worms were stimulated by tapping the plate and immediately recorded for 200 cycles

at room temperature using a Leica (Amsterdam, The Netherlands) M205 FA fluorescent microscope and Leica DFC 365 FX camera.

Images were captured using Leica Application Suite X software and then processed with the wrMTrck plugin for ImageJ to get the average moving speed of each worm. Then the median moving speed of worms in every condition is used to calculate the difference of moving speed between conditions. $Log_2$((median speed of condition 1) − (median speed of condition 2)) is used to calculate the difference between the two conditions. For instance, $log_2$(median speed of *mrps-5* RNAi worms − median speed of HT115 fed worms) to measure how much *mrps-5* RNAi treatment changed the moving speed compared to control.

To screen which candidate genes may be responsible for the healthspan improved by mitochondrial translation inhibition, *E. coli* HT115 was used as RNAi control. For single RNAi experiments, targeted RNAi bacteria were mixed with *E. coli* HT115 in a 1:1 ratio. For double RNAi experiments, bacterial cultures were mixed in a 1:1 ratio.

### Lifespan assay
Lifespan assays were performed as previously described[6,16]. In brief, worms were grown on NGM plates seeded with *E. coli* OP50 until reaching the L4 stage and then transferred to RNAi plates (F0). F0 adults were synchronized by hypochlorite treatment after 24 h, then transferred to NGMi RNAi plates to generate F1 offspring. At the L4 stage, a total of around 100 F1 worms per condition were transferred to RNAi bacteria-seeded plates containing 10 µM 5-FU to prevent the growth of progeny. Worms were transferred to fresh RNAi plates once a week, and after two transfers, no 5-FU was added to the plates. HT115 RNAi bacteria were used as the RNAi control. For single *mrps-5* RNAi, 20% of *mrps-5* RNAi bacteria was mixed with 80% HT115. For single *cco-1* RNAi, 20% of *cco-1* RNAi bacteria was mixed with 80% HT115. For single candidate RNAi (*C32E8.9, sta-2, nsy-1, sta-2, sma-3, sma-4*), 80% of the candidate RNAi bacteria was mixed with 20% HT115. For double RNAi, 80% of the candidate RNAi bacteria was mixed with 20% *mrps-5* RNAi bacteria. Worms were examined every other day by prodding with a platinum wire. All lifespan assays were performed at least twice, one of which is represented in the data shown. Statistical analyses of lifespan were calculated by Log-rank (Mantel−Cox) tests on Kaplan−Meier curves in GraphPad Prism.

### PA01 slow-killing assays
Slow-killing assays with *Pseudomonas aeruginosa* strain PA01 were performed as previously described with minor modifications[52]. PA01 was generously provided by the lab of Prof. Dr. S. (Stanley) Brul at the University of Amsterdam. PA01 was cultured in LB media at 37 °C overnight, and 5 µL of the liquid culture was then seeded on the center of high-peptone NGM plates (0.35% bactopeptone). The PA01-seeded plates were incubated at 37 °C for 24 h and kept at room temperature for 8–24 h before assays. N2 worms were pretreated with RNAi from the parental stage (same as in the lifespan assay). Then, around 100 F1 L4 stage worms were transferred to the PA01-seeded plates supplemented with 5FU. The animals were incubated at 25 °C, scored every day, and counted as dead if they did not respond to prodding. All slow-killing assays were performed at least twice, one of which is represented in the data shown. Statistical analyses of lifespan were calculated by Log-rank (Mantel−Cox) tests on Kaplan−Meier curves in GraphPad Prism.

### RNA isolation and quantitative RT-PCR
RNA extraction and qRT-PCR were performed as previously described with minor modifications[16]. Total RNA from worms was isolated using TRI reagent (Sigma-Aldrich). 1 µg of extracted RNA was reverse transcribed into cDNA using the QuantiTect Reverse Transcription Kit (QIAGEN; Venlo, The Netherlands). qPCR was conducted using the LightCycler® 480 SYBR Green I Master (Roche; Woerden, The

Netherlands) and measured with the LightCycler® 480 Instrument II (Roche). Relative quantifications were normalized to the reference genes *tba-1* and *F35G12.2*. For each condition, more than three independent samples were prepared. All experiments were performed at least twice. A Student's *t*-test was used to compare the differences in gene expression between different conditions. Data visualization was conducted using GraphPad Prism.

### Microscopy analysis
Animals were immobilized on 3 cm NGM plates in 5 mM levamisole. Images were captured using a Leica M205 FA fluorescent microscope and a Leica DFC 365 FX camera (the same machine used in the above-mentioned RNAi screening). The area of a worm body was selected and quantified using ImageJ. For each replicate, approximately 5 randomly chosen worms were recorded, and the average body size or body width was calculated to represent each replicate. A *t*-test was used to compare the differences in body size or body width between different conditions. Data visualization was conducted using GraphPad Prism.

### Fluorescent microscopy analysis
GFP expression and quantification were carried out as described previously[6]. Briefly, around eighty *hsp-6p::gfp* worms (day 1 adults) were mounted on 2% agarose pads in 10 mM tetramisole (Sigma) and examined using a Leica M205 FA fluorescent microscope. The GFP fluorescence was quantified by ImageJ. Experiments were conducted with worms from three different plates. Each experiment was repeated at least twice.

### Measurement of bacterial avoidance behavior
Bacteria avoidance assays were performed as previously described[36]. In brief, 100 µL of RNAi bacteria were seeded on the center of NGMi plates, allowing at least one day to dry at room temperature. Then plates with uniform circular lawns were selected for the following experiments. On the next day, around 100 L1 animals were transferred to the center of the bacterial lawn without disturbing or spreading the lawn. Each animal was scored as inside or outside the bacteria (avoidance percent = $N_{out}/N_{total} * 100$) at the late L4 stage.

### Selection of candidate genes for mitochondrial translation inhibition
The candidate list of genes mediating mitochondrial translation inhibition was selected based on gene and protein expression from Molenaars et al., available as supplemental data therein[16]. Three datasets were used to establish a common gene product expression signature. (1) Gene expression from worms treated with *mrps-5* RNAi compared to control, (2) protein expression from worms treated with *mrps-5* RNAi compared to control, and (3) gene expression from mouse livers treated with doxycycline compared to control. For all three datasets, a cutoff was selected based on a Variable Importance in Projection (VIP) score >1, and a fold change >0, whereby the VIP score estimates the importance of each gene product in a PLS-DA model (as originally described in Molenaars et al. 2020) and the fold change was defined to select upregulated genes. The overlap between all three datasets produced a final candidate gene list of 19 upregulated genes resulting from mitochondrial translation inhibition.

### RNA sequencing
RNA sequencing was carried out as described[53] with minor modifications. N2 worms were synchronized and treated with RNAi bacteria as described from the L1 stage. For RNA sequencing, day 5 adult worms were selected, as they exhibited a more pronounced phenotype in the mobility assay observed at later stages. Additionally, this time point was chosen to minimize potential confounding effects associated with egg production. Day 5 adult animals were harvested by washing three times with M9 buffer and two times with water before being snap-

frozen in liquid nitrogen. ~1000 worms for each sample. Total RNA was extracted as described above in the "RNA Isolation" section. Contaminating genomic DNA was removed using RNaseFree DNase (QIAGEN). RNA was quantified with a NanoDrop 2000 spectrophotometer (Thermo Scientific; Breda, The Netherlands). The qualities of RNA samples were checked with TapeStation (Agilent, CA, USA). Sequencing libraries were constructed by using KAPA mRNA HyperPrep Kit (Roche, Switzerland) and paired-end sequencing of Illumina NovaSeq was performed (Macrogen, Seoul, South Korea).

Reads were subjected to quality control FastQC and trimmed using fastp (version 0.23.2)[54] and aligned to the *C. elegans* genome obtained from Ensembl (wbcel235), using STAR2 (version 2.5.4)[55]. The STAR gene-counts for each alignment were analyzed for differentially expressed genes using the R package DESeq2 (version 1.32.0)[56] using a generalized linear model. Variance Stabilizing Transformation data generated by DESeq2 was used for principal component analyses to explore the primary variation in the data. Count data were normalized to counts per million (CPM) using edgeR (version 3.36.0)[57]. Biological process (BP) over-representation analysis and Gene Set Enrichment Analysis (GSEA) were performed using Clusterprofiler (version 4.0.5)[58] and org.Ce.eg.db (version 3.13.0). ggplot2 (version 3.4.2) was used to generate heatmaps and various figures. Code was executed in R version 4.1.1.

### One-phase lipidomic extraction and lipidomics in *C. elegans*

The extraction and analysis of lipids were performed as previously described[59,60]. Briefly, worms were synchronized at L1 and subjected to specified RNAi bacteria until reaching the day-5 adult stage. 2000 worms per sample were collected for lipid extraction. Lipidomics analysis was performed as described[61]. The HPLC system consisted of an Ultimate 3000 binary HPLC pump, a vacuum degasser, a column temperature controller, and an auto sampler (Thermo Scientific, Waltham, MA, USA). The column temperature was maintained at 25 °C. The lipid extract was injected onto a "normal phase column" LiChrospher 2 × 250-mm silica-60 column, 5 μm particle diameter (Merck, Darmstadt, Germany) and a "reverse phase column" Acquity UPLC HSS T3, 1.8 μm particle diameter (Waters, Milford, Massachusetts, USA). A Q Exactive Plus Orbitrap (Thermo Scientific) mass spectrometer was used in the negative and positive electrospray ionization mode. In both ionization modes, mass spectra of the lipid species were obtained by continuous scanning from m/z 150 to m/z 2000 with a resolution of 280,000 full width at half maximum (FWHM).

Similar differential analysis was performed using the Bioconductor package limma version 3.58[62], with a generalized linear model. Results of the statistical tests were corrected for multiple testing using the Benjamini-Hochberg method. The 'GSEA' function from the clusterProfiler R package[58] was used to conduct lipid class enrichment analysis (Fig. 6d), which is designed to accept customized annotations. This enrichment analysis utilized lipid classes for lipid annotations.

### Statistics and reproducibility

All the assays were conducted at least twice independently, and the statistical analysis used in this study is described in the figure legends and/or methods. No statistical method was used to predetermine the sample size. Comparison between more than two groups was assessed by using a One-way ANOVA test. Prism 9 (GraphPad Software) was used for statistical analysis of all lifespan, qRT-PCR, and slow-killing assay experiments. ****$p < 0.0001$; ***$p < 0.001$; **$p < 0.01$; *$p < 0.05$; n.s., not significant.

### Reporting summary

Further information on research design is available in the Nature Portfolio Reporting Summary linked to this article.

## Data availability

The mRNA-Seq data generated in this study have been deposited in the GEO database under the accession number "GSE248642" (https://www.ncbi.xyz/geo/query/acc.cgi?acc=GSE248642). The lipidomes generated in this study are provided in the Source Data. Source data are provided with this paper.

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

## Acknowledgements

Work in the Houtkooper group is financially supported by an ERC Starting grant (no. 638290), a VIDI grant from ZonMw (no. 91715305) and by the Velux Stiftung (no. 1063). G.E.J. is supported by a VENI grant from ZonMw, and AGEM Talent and Development grants. I.M.H. is supported by a China Scholarship Council grant. A.W.G. was supported by

Amsterdam UMC Postdoc Career Bridging Grant, Horizon-MSCA-PF-EF-2022 (101108082), and AGEM Talent Development Grant (2023). Work in the Bommer group is supported by an ERC Consolidator grant (no. 771704) and the WELRI institute. We thank the Caenorhabditis Genetics Center (CGC) at the University of Minnesota for providing *C. elegans* strains, which is funded by the NIH Office of Research Infrastructure Programs (P40 OD010440). We acknowledge the use of Biorender to generate Fig. 4a and part of Figs. 1a, 3a, S3b.

## Author contributions

I.M.H., R.H.H., G.E.J. and M. Molenaars conceived and designed the project. I.M.H., M. Molenaars, M. Modder, and A.B. performed experiments. I.M.H. and G.E.J. analyzed the data. I.M.H. performed bioinformatics analyses of the RNA sequencing data. B.V.S., Y.J. and M.v.W. performed, analyzed, and aided in the interpretation of lipidomic data. J.P.D. and G.T.B. provided critical advice. I.M.H., G.E.J., A.W.G. and R.H.H. wrote the manuscript with contributions from all other authors.

## Competing interests

The authors declare no competing interests.
