## [Transparent Peer Review file · Nature Communications]

Immuno-metabolic stress responses control longevity from mitochondrial translation inhibition in *C. elegans*

Corresponding Author: Professor Riekelt Houtkooper

Version 0:

Reviewer comments:

Reviewer #1

(Remarks to the Author)

This manuscript explored the downstream of mitochondrial translation other than UPRmt. The authors discovered that C32E8.9-controlled lipid metabolism and TGF- β -controlled immune response could be critical in the longevity induced by suppressing mitochondrial translation.

Given the pivotal role of mitochondria in cellular metabolism, suppressing mitochondrial translation or any other interference with mitochondrial function should have a complex effect on longevity. It is nice to see a study on other potential downstream besides the extensively studied UPRmt. The authors also found that C32E8.9 does not regulate UPRmt and functions independently of it, highlighting their discoveries as a new piece in this puzzle game. Therefore, this manuscript is quite interesting and informative to the ageing researchers.

However, this study still has some obvious shortcomings, which should be carefully addressed.

1. The biggest issue is the methodology of this study. The authors performed all their analyses solely by RNAi. That raises two concerns:

a. RNAi is a convenient but not the only or best way to interfere with gene expression. Although a genetic mutant of C32E8.9 is not yet available in a public source (e.g., CGC), the authors can easily make a knock-out mutant with CRISPR/Cas9 technology. If such a mutant is unavailable, for example, because C32E8.9 is essential to embryonic development, the authors should also report this in the manuscript to better understand this gene's function.

The authors studied C32E8.9 because it is upregulated upon RNAi against mrps-5. Then an obvious assay is to check whether overexpressing C32E8.9 in wild-type worms could phenocopy or at least partially phenocopy mrps-5 RNAi. As lifespan is an integrated result from various causes, the phenocopy is not necessarily in a change of survival curve but could also be in immune response (PA01 resistance) or lipidomic changes.

The strain overexpressing C32E8.9 also provides a chance to examine the genetic epistasis between this gene and sma-4 (see issue '2').

Taken together, another independent method to misregulate C32E8.9 is crucial to consolidate the main findings of this manuscript.

b. When describing 'Lifespan assay' in the 'Methods' section, the authors carefully mentioned the settings for each RNAi assay, especially that the concentration of mrps-5 RNAi bacteria is the same in its single RNAi and double RNAi with other candidates. Do these settings apply to all RNAi assays in this manuscript? This issue is critical to interpreting the RNAi efficiency, especially in double RNAi assays.

2. The interaction between sma-4 and C32E8.9.

The authors proposed that sma-4 or TGF- β signalling functions downstream of C32E8.9. The most direct evidence to support this critical speculation is the alterations of these genes expression in RNA-Seq analysis (Fig. 4b). But that is far from enough to establish this crucial conclusion in this manuscript. The authors first need to confirm the change of these genes expression by another method, such as qPCR.

Moreover, the regulation in expression does not necessarily indicate a functional interaction. The authors also showed that inhibiting sma-4 suppresses a series of mrps-5 RNAi-induced phenotypes. Yet, these additional results can only indicate that sma-4 works downstream of mrps-5. It can either function under or in parallel with C32E8.9. Therefore, it is essential to further examine the genetic interaction between sma-4 and C32E8.9. For example, if overexpressing C32E8.9 has a similar phenotype as mrps-5 RNAi, mutating or suppressing sma-4 should block this phenotype. Or, arbitrarily overexpressing sma-

4 may more or less inhibit the effect induced by C32E8.9 RNAi.

3. C32E8.9 is an enzyme regulating fatty acids metabolism. One of the highlights of this manuscript is to pinpoint fatty acids metabolism as a critical downstream of mitochondrial translation. Unlike UPRmt, which is easily linked with the disruption of protein homeostasis in mitochondria, how fatty acids and mitochondria interact with each other needs to be further clarified in this manuscript.

a. How does mrps-5 RNAi upregulate C32E8.9? Since the authors have the transcriptomic changes upon mrps-5 RNAi, it is at least feasible to deduce which transcription factor (TF) controls the upregulation of C32E8.9. It would be great, but not essential, if the authors could further prove this TF-mediated control.

b. In Fig. 6c, the authors showed no significantly altered lipids upon mrps-5 RNAi. However, the subsequent Figs. 6d and e showed that mrps-5 controls a few lipid classes through C32E8.9. It is likely that many lipids in these classes (e.g., TG) are changed upon mrps-5 RNAi without statistical significance. To better present the role of lipids remodeling in mitochondrial translation-induced longevity, the manuscript could include another supplementary figure showing:

b1. A similar figure panel as Fig. 6c but with a threshold of ' $\log_2FC > 1$ ' only.

b2. Figure panels using Venn diagrams to compare the changes of lipids, all and those of TG and 2-acyl LPC, upon mrps-5 RNAi, C32E8.9 RNAi, and double RNAi against mrps-5 and C32E8.9.

4. Other minor issues:

a. It would be nice to show the scale of \log_2FC in Fig. 1b, as done in Fig. 1c.

b. Why did the authors choose day 5 adult worms for RNA-Sequencing (page 6, line 169)?

c. A supplementary table summarizing the results of all lifespan and PA01 slow-killing assays should be included in the manuscript, showing:

c1. The number of tested worms.

c2. The number of biological replicates.

c3. Average and maximum lifespan.

c4. P values from Log-rank tests.

Reviewer #2

(Remarks to the Author)

In this manuscript by Houtkooper and colleagues, the authors detail the use of a multi-omics strategy to identify additional response elements beyond the UPRmt that are responsible for lifespan and healthspan extension in response to inhibition of mitochondrial protein translation.

One problem with the use of dual RNAi, which is used in several places in this paper, both in the screen in Fig. 1, in Figs. 1D-E, Fig1G, figure 2A/D, Figure 3 RNAseq, etc.... One trivial explanation for the apparent suppression of mrps-5 phenotypes with concomitant C32E8.9 RNAi is that neither RNAi is as efficient. This should be ruled out by quality controlling RNAi efficiency in dual vs. single knockdown. This is incredibly important in quality control for many of the analyses performed in the manuscript. Strong consideration should be given to depleting C32E8.9 by other means such as degen-based technology, availability of which is widespread. Are mutants of C32E8.9 available? Viable?

In general the statistical tests (and multiple hypothesis correction) used are only spottily indicated in the figures, e.g. missing in 1e, 1g. In figure 2, the statistical test used is frankly inappropriate (t-test comparisons are not appropriate across multiple groups (ANOVA or similar should be used) and these tests require multiple hypothesis correction). The same is true for figure 4J, 5E, 5F. This calls some of the conclusions into question and these issues must be addressed to establish validity of the results presented. In general the results should be gone through, tests should be specified, and appropriate multiple hypothesis testing correction conducted where appropriate, and t-test generally only used for data with two groups with only a pairwise comparison.

Echoing the concern above, double RNAi is not a preferred method when viable loss of function mutants are available, as in the case of sma-3 and sma-4. Given that these are key conclusions made by the manuscript, loss of function mutants should be used to substantiate data with less rigorous double RNAi.

It does not appear that the data substantiate the claim that "This protective effect was blunted upon double knockdown of mrps-5 and C32E8.9 (Fig. 5c). We observed a similar pattern with double knockdown of mrps-5 and sma-4 (Fig. 5d)." The relationship looks more parallel with sma-4 than epistatic as with C32E8.9.

The conclusion from figure 6B that the lipidome does not change with mrps-5 RNAi suggests that in fact remodeling of the lipidome (at least as can be assessed with static measurements such as LCMS lipidomics) is not involved in the lifespan extension of mrps-5 knockdown. This is misleading, and perhaps the enrichment analysis done for classes of lipids as in figure 6D should be showcased first and the point made more strongly with these data. However, the conclusion that TG, particularly those increased in chain length and desaturation, which should be decreased in abundance with activation of fatty acid catabolism, is puzzling and does not immediately follow from the data presented. The conclusion that the remodeling of the lipidome is linked to the "activation" of C32E8.9 expression and the shift towards longer, more desaturated fatty acids could be explicitly tested with targeted experiments. The authors should address the possibility that genes encoding enzymes involved in fatty acid elongation or desaturation act mechanistically downstream of mrps-5 and/or C32E8.9.

Minor comments:

Figure 2 is all negative data essentially and could be re-purposed to the supplement.

There is a fair amount of negative data (e.g. 5A) presented before more meaningful data and this detracts from the message presented.

Reviewer #3

(Remarks to the Author)

Inhibition of mitochondrial translation results in lifespan extension, but it is not understood the molecular basis for this increase in longevity. To address this problem, the authors analyze RNA-seq data and take a candidate approach to identify genes that alleviate the beneficial effects of knocking down *mrps-5*. Using a dual RNAi approach the authors show that knockdown of C32E8.9 reverses many of the effects of *mrps-5* knockdown including lifespan extension, small body size, and lipidomic changes. Knockdown of *mrps-5* also causes an immune response that is absent when C32E8.9 is concurrently knocked down. In addition, the authors find that knockdown of the transcription factor *sma-4* also reverses effects caused by *mrps-5* knockdown. Together, the authors suggest that C32E8.9 is a necessary factor for mediating the impact of mitochondrial translation inhibition.

Main comment:

1. Almost all of the authors' results relies on observations that double RNAi knockdown of *mrps-5* with either C32E8.9 or *sma-4* results in alleviation of various *mrps-5* phenotypes. The authors interpret these results as C32E8.9 or *sma-4* being necessary for the effects caused by the *mrps-5* knockdown. An alternative interpretation of this data is that dual feeding of RNAi bacteria is causing less efficient knockdown of *mrps-5*. This phenomenon has been previously observed in *C. elegans* and reported in several publications (doi: 10.1186/gb-2000-2-1-research0002, DOI: 10.2144/000113365). The authors need to use an alternative to dual RNAi knockdowns to show these effects. Mutant alleles exist for *sma-4*, and although C32E8.9 and *mrps-5* mutant alleles are not available from the strain The Caenorhabditis Genetics Center, generating mutants in *C. elegans* is now straightforward. The authors should minimally repeat the phenotypic experiments in genetic mutants and ideally would also confirm their transcriptomic and metabolomic data with these mutants. Although having genetic mutants of C32E8.9 will be necessary to full support the role of C32E8.9 in suppressing the effects of mitochondrial translation inhibition, the authors could also test if C32E8.9 RNAi knockdown by itself reverses the phenotypes caused by doxycycline inhibition of mitochondrial translation.

Minor comment:

1. In Figure 1b there are no units to the scale.

Version 1:

Reviewer comments:

Reviewer #1

(Remarks to the Author)

The authors have effectively addressed all my concerns, and the manuscript is suitable for publication in Nature Communications after the following minor textual corrections are made:

1. Line 66, page 3: *C. elegans* should be italicized.
2. Lines 145–146, page 6: Supplementary Fig. 1f should be cited to demonstrate the efficiency of the C32E8.9 knockout.
3. In the legend of Supplementary Fig. 3i, please include the genotype of the C32E8.9 overexpression strain to help readers understand the source of the fluorescent signals.
4. Please provide a brief explanation for the use of the *dpy-30* promoter to overexpress C32E8.9 and clarify why the GFP signal is predominantly observed in the pharynx. Could the strong GFP signal result from leakage from the strong *Pmyo-2::mCherry*?

These minor revisions will enhance the clarity and completeness of the manuscript.

Reviewer #2

(Remarks to the Author)

The authors have done a suitable job of addressing my concerns, most of which were methodological and statistical. The manuscript is interesting and adds new dimensions to our understanding of stress responses in aging and will be of interest to a broad readership.

Reviewer #3

(Remarks to the Author)

The inclusion of the C32E8.9 mutant data greatly strengthens the manuscript, but although the authors mostly address my concerns, there are still several issues that should be addressed.

Major points

1. Although the inclusion of mutants now alleviates the concern that the phenotypes observed were due to a reduction in RNAi efficiency in the double knockdown condition, the data included by the authors in S1C, appears to show a reduction of C32E8.9 RNAi efficiency, albeit not significant to $p < .05$. I feel that authors should mention this as a potential caveat to their study in the discussion.
2. The authors state in their reporting summary that all data was replicated at least twice, but table s1 shows that several of the lifespans in figure 4 only have a single replicate. Most of the figures legends do not say how many animals were used for each experiments and how many biological replicates there were. This information needs to be included in the figure legends and in the reporting summary. The individual points should be shown for the survival curves. All the survival curve data should be shown, either on the same graph, or in the supplementary.
3. The data availability statement for the paper says there is a source data file, but that does not appear to have been included or not made available to the reviewers.

Minor points

1. Figure S1. The graphs are missing units. Several of the graphs list "dox" as an RNAi condition. I believe it should be ctl +dox.
2. Figure 2d. Scale bars should be included.

Version 2:

Reviewer comments:

Reviewer #1

(Remarks to the Author)

The authors have addressed all my remaining concerns.

Reviewer #3

(Remarks to the Author)

the authors have addressed all my concerns and I now enthusiastically support publication of their manuscript

Reviewer #1: This manuscript explored the downstream of mitochondrial translation other than UPR^{mt}. The authors discovered that *C32E8.9*-controlled lipid metabolism and TGF- β -controlled immune response could be critical in the longevity induced by suppressing mitochondrial translation. Given the pivotal role of mitochondria in cellular metabolism, suppressing mitochondrial translation or any other interference with mitochondrial function should have a complex effect on longevity. It is nice to see a study on other potential downstream besides the extensively studied UPR^{mt}. The authors also found that *C32E8.9* does not regulate UPR^{mt} and functions independently of it, highlighting their discoveries as a new piece in this puzzle game. Therefore, this manuscript is quite interesting and informative to the ageing researchers.

We thank the reviewer for the kind words and are pleased that the reviewer considers our work novel, informative and valuable for the aging research community. Below we explain how we have addressed the reviewer's concerns point by point.

However, this study still has some obvious shortcomings, which should be carefully addressed. 1. The biggest issue is the methodology of this study. The authors performed all their analyses solely by RNAi. That raises two concerns:

a. RNAi is a convenient but not the only or best way to interfere with gene expression. Although a genetic mutant of *C32E8.9* is not yet available in a public source (e.g., CGC), the authors can easily make a knock-out mutant with CRISPR/Cas9 technology. If such a mutant is unavailable, for example, because *C32E8.9* is essential to embryonic development, the authors should also report this in the manuscript to better understand this gene's function.

We agree that relying solely on RNAi has limitations, such as potential off-target effects and incomplete knockdown. To address this concern, we successfully generated a *C32E8.9* knockout strain using CRISPR-Cas9 technology. The *C32E8.9* knockout strain was both viable and fertile under laboratory conditions, indicating that *C32E8.9* is not essential for embryonic development. Details regarding the design, construction, and validation of this mutant have been added to the Methods (page 16) and in Supplementary Figure 1f.

To validate the role of *C32E8.9*, we examined the lifespan of *mrps-5* RNAi-treated *C32E8.9* knockout worms in comparison to control strains. Our results show that the lifespan extension induced by *mrps-5* RNAi was abolished in the *C32E8.9* knockout strain, confirming that *C32E8.9* is indeed required for this beneficial effect. We also observed that the lifespan of the *C32E8.9* knockout strain itself did not significantly differ from that of the wild-type controls. These findings are now added in page 5-6 and included in Figure 1f. Additionally, we performed a PA01 resistance assay under the same experimental conditions. In line with our lifespan findings, the enhanced survival observed with *mrps-5* RNAi after PA01 exposure was absent in the *C32E8.9* knockout strain. These data are detailed in page 9-10 and shown in Figure 5c.

This data collectively strengthens our conclusion that *C32E8.9* is required for the beneficial effect conferred by *mrps-5* RNAi.

The authors studied *C32E8.9* because it is upregulated upon RNAi against *mrps-5*. Then an obvious assay is to check whether overexpressing *C32E8.9* in wild-type worms could phenocopy or at least partially phenocopy *mrps-5* RNAi. As lifespan is an integrated result from various causes, the phenocopy is not necessarily in a change of survival curve but could also be in immune response (PA01 resistance) or lipidomic changes. The strain overexpressing *C32E8.9* also provides a chance to examine the genetic epistasis between this gene and *sma-4* (see issue '2'). Taken together, another independent method to misregulate *C32E8.9* is crucial to consolidate the main findings of this manuscript.

Building on your insightful suggestion, we generated a *C32E8.9* overexpression (OE) strain to determine whether increased expression of this gene replicates the beneficial effects associated with *mrps-5* RNAi. The details of the strain construction are described in the Methods (page 17), and confirmation experiments of the OE strain are included in Supplementary Figure 3i.

We examined the lifespan of the OE strain in comparison to controls with the same genetic background. Our results indicate that the *C32E8.9* OE strain exhibited a significant extension in lifespan, consistent with the effects observed with *mrps-5* RNAi treatment, though the lifespan extension is not as strong as we observed in *mrps-5* RNAi. These findings are presented in page 11 and Figure 7a.

Beyond lifespan, we also evaluated whether the overexpression of *C32E8.9* enhances immune function by performing a PA01 resistance assay. Consistent with the lifespan results, we observed that *C32E8.9* OE worms displayed increased survival compared to controls. These results have been added to page 11, Figure 7b.

Overall, the data obtained from the *C32E8.9* overexpression experiments support the functional role of this gene in mediating the beneficial effects initially observed with *mrps-5* RNAi.

b. When describing 'Lifespan assay' in the 'Methods' section, the authors carefully mentioned the settings for each RNAi assay, especially that the concentration of *mrps-5* RNAi bacteria is the same in its single RNAi and double RNAi with other candidates. Do these settings apply to all RNAi assays in this manuscript? This issue is critical to interpreting the RNAi efficiency, especially in double RNAi assays.

We confirm that the bacterial concentration for *mrps-5* RNAi was consistent across both single and double RNAi assays, and this approach was applied throughout the study. In response to your suggestion, we have added a table detailing the concentration of RNAi bacteria used for each gene in the experiments, which can be found in the Supplementary Table 1-2.

To confirm that the knockdown efficiency was not compromised in the double RNAi setup, we conducted qPCR analyses for both *mrps-5* and *C32E8.9*. We compared their expression levels under single RNAi versus double RNAi conditions. For *mrps-5*, no significant difference in expression was observed, confirming consistent knockdown efficiency (page 5 and Supplementary Figure 1d). Similarly, *C32E8.9* expression remained unaffected in the double RNAi setup, as shown in page 6 and Supplementary Figure 1b-c.

These findings validate the effectiveness of our RNAi approach across both single and double RNAi conditions.

2. The interaction between *sma-4* and *C32E8.9*. The authors proposed that *sma-4* or TGF- β signaling functions downstream of *C32E8.9*. The most direct evidence to support this critical speculation is the alterations of these genes expression in RNA-Seq analysis (Fig. 4b). But that is far from enough to establish this crucial conclusion in this manuscript. The authors first need to confirm the change of these genes expression by another method, such as qPCR.

Thank you for the suggestion. In response, we performed qPCR analysis of *sma-4* under the same conditions as our RNA-Seq experiments. The qPCR data showed that *sma-4* expression was upregulated by *mrps-5* RNAi but decreased in the double RNAi with *C32E8.9*, consistent with the RNA-Seq results. These findings have been added to page 9 and Supplementary Figure 1e.

Moreover, the regulation in expression does not necessarily indicate a functional interaction. The authors also showed that inhibiting *sma-4* suppresses a series of *mrps-5* RNAi-induced

phenotypes. Yet, these additional results can only indicate that *sma-4* works downstream of *mrps-5*. It can either function under or in parallel with *C32E8.9*. Therefore, it is essential to further examine the genetic interaction between *sma-4* and *C32E8.9*. For example, if overexpressing *C32E8.9* has a similar phenotype as *mrps-5* RNAi, mutating or suppressing *sma-4* should block this phenotype. Or, arbitrarily overexpressing *sma-4* may more or less inhibit the effect induced by *C32E8.9* RNAi.

Our qPCR and RNA-seq data demonstrated that *C32E8.9* RNAi reverses the upregulated mRNA expression of *sma-4* induced by *mrps-5* RNAi. While these findings suggest that *sma-4* may act downstream of *C32E8.9* following mitochondrial translation inhibition, it remains uncertain whether *sma-4* functions downstream of or in parallel with *C32E8.9*. Because these experiments do not formally indicate a functional interaction, we have toned down our specific conclusions on *sma-4* and accordingly revised the manuscript to address this point and acknowledge the limitations of our study (see page 14).

3. *C32E8.9* is an enzyme regulating fatty acids metabolism. One of the highlights of this manuscript is to pinpoint fatty acids metabolism as a critical downstream of mitochondrial translation. Unlike UPR^{mt}, which is easily linked with the disruption of protein homeostasis in mitochondria, how fatty acids and mitochondria interact with each other needs to be further clarified in this manuscript.

We agree that the role of *C32E8.9* in regulating fatty acid metabolism is an intriguing aspect of our findings. We acknowledge that lipid metabolism is complex. The diverse lipid configurations (varying in chain length, saturation, and bond positioning), combined with the interplay of multiple, often promiscuous, enzymes, makes it challenging to pinpoint specific metabolic changes directly attributable to *C32E8.9*'s role. To address the complexity and its potential role in the *mrps-5/C32E8.9* longevity axis, we revised the discussion section of our manuscript (page 15). While our current manuscript may not solve this quest, it is evident that inhibiting mitochondrial translation has beneficial effects on longevity that are associated with lipid changes and hence identifying the specific lipid modifications driving these effects remains of interest in our future work. We also discuss the possibility of other potential effectors of UPR^{mt} and stress responses influenced by *C32E8.9*. By addressing these limitations, our future studies will provide a more comprehensive understanding of the mechanisms by which mitochondrial function influences lipid metabolism and aging.

a. How does *mrps-5* RNAi upregulate *C32E8.9*? Since the authors have the transcriptomic changes upon *mrps-5* RNAi, it is at least feasible to deduce which transcription factor (TF) controls the upregulation of *C32E8.9*. It would be great, but not essential, if the authors could further prove this TF-mediated control.

To explore whether/which TFs might contribute to the upregulation of *C32E8.9* upon *mrps-5* RNAi, we conducted a bioinformatics analysis focusing on identifying potential TFs that could be involved in this regulation.

We analyzed the promoter region of *C32E8.9* (2 KB upstream of the transcription start site, chr1:3,798,069-3,800,068) using the JASPAR database [<https://jaspar.elixir.no/>], a widely used and reliable resource for DNA-binding motif prediction. By applying a 90% threshold for relative sensitivity and specificity, we identified 55 transcription factors that have a high probability of binding to this promoter region as indicated in Reviewer Figure 1a.

Next, we compared these predictions with our transcriptomic data from *mrps-5* RNAi-treated samples, revealing that 18 of these TFs are upregulated by *mrps-5* RNAi. These TFs may bind to the *C32E8.9* promoter and contribute to its upregulation under these conditions (illustrated in Reviewer Figure 1b).

Although we have identified promising TF candidates, we acknowledge that bioinformatic predictions alone cannot confirm their regulatory roles. Further validation through ChIP-seq

and luciferase assays would provide stronger evidence of their contribution to *C32E8.9* regulation. However, we feel that these are beyond the scope of the current manuscript.

Reviewer Figure 1 | Potential transcription factors controlling the upregulation of *C32E8.9* following *mrps-5* RNAi. **a** | Venn diagram showing the overlap between transcription factors predicted from JASPAR and upregulated genes identified in response to *mrps-5* RNAi. **b** | Heatmap of mRNA expression of 18 transcription factors in four RNAi knockdown conditions. Fill color represents the Z-score transformed counts per million (CPM) gene expression for each gene across all samples. For visualization purposes, values >2 were capped at 2, while values <-2 were capped at -2.

b. In Fig. 6c, the authors showed no significantly altered lipids upon *mrps-5* RNAi. However, the subsequent Figs. 6d and e showed that *mrps-5* controls a few lipid classes through *C32E8.9*. It is likely that many lipids in these classes (e.g., TG) are changed upon *mrps-5* RNAi without statistical significance. To better present the role of lipids remodelling in mitochondrial translation-induced longevity, the manuscript could include another supplementary figure showing:

b1. A similar figure panel as Fig. 6c but with a threshold of ' $\log_2FC > 1$ ' only.

Changes have been made as suggested. We have included the counts of lipids using the threshold of ' $\log_2FC > 1$ ' in new Supplemental Figure 3d. The single knockdown of *C32E8.9* showed the strongest effect, with 195 lipids accumulating and 361 depleted, compared to 4 accumulating and 10 depleted in *mrps-5* RNAi, consistent with the PCA results.

b2. Figure panels using Venn diagrams to compare the changes of lipids, all and those of TG and 2-acyl LPC, upon *mrps-5* RNAi, *C32E8.9* RNAi, and double RNAi against *mrps-5* and *C32E8.9*.

We have generated the Venn diagrams as requested and visualized them using an UpSetR plot in Supplemental Figure 3f-h. Panel 3f shows the distribution of all tested lipids, while panels 3g and 3h focus on TGs and 2-acyl LPC, respectively. These plots highlight the substantial impact of *C32E8.9* knockdown, with the largest set size of both upregulated and downregulated lipids. The intersections illustrate the overlap in lipid changes between single and double knockdowns, providing insights into shared and unique lipid alterations.

4. Other minor issues:

a. It would be nice to show the scale of \log_2FC in Fig. 1b, as done in Fig. 1c.

We have added separate scales for mRNA and protein in *C. elegans* and mRNA in mice in Figure 1b to ensure consistency and improve clarity.

b. Why did the authors choose day 5 adult worms for RNA-Sequencing (page 6, line 169)?

We selected day 5 adult worms for RNA sequencing because at later age there was a stronger phenotype in the mobility assay which we wanted to avoid. Moreover, we wanted to avoid potential confounding effects from egg production. This explanation has been added in the Methods-RNA sequencing page 20.

c. A supplementary table summarizing the results of all lifespan and PA01 slow-killing assays should be included in the manuscript, showing:

- c1. The number of tested worms.
- c2. The number of biological replicates.
- c3. Average and maximum lifespan.
- c4. P values from Log-rank tests.

We have added Supplementary Table 1, which includes the requested details for the lifespan and Supplementary Table 2 for PA01 slow-killing assays.

Reviewer #2 (Remarks to the Author): In this manuscript by Houtkooper and colleagues, the authors detail the use of a multi-omics strategy to identify additional response elements beyond the UPR^{mt} that are responsible for lifespan and healthspan extension in response to inhibition of mitochondrial protein translation.

Thank you for your insightful comments. We have made revisions to address these concerns point by point.

One problem with the use of dual RNAi, which is used in several places in this paper, both in the screen in Fig. 1, in Figs. 1D-E, Fig1G, figure 2A/D, Figure 3 RNAseq, etc.... One trivial explanation for the apparent suppression of *mrps-5* phenotypes with concomitant *C32E8.9* RNAi is that neither RNAi is as efficient. This should be ruled out by quality controlling RNAi efficiency in dual vs. single knockdown.

We fully understand the question regarding the efficiency of RNAi in dual knockdown experiments and the potential impact this could have on our results. To address this, we measured the expression levels of *mrps-5* using qPCR in both single *mrps-5* RNAi and double *mrps-5* + *C32E8.9* RNAi conditions. Our qPCR analysis revealed that the expression of *mrps-5* was effectively suppressed in both cases, with no significant difference between the single and dual RNAi treatments. This confirms that the efficiency of *mrps-5* knockdown is not compromised when combined with *C32E8.9* RNAi. The results of this analysis are presented in page 5 and Supplementary Figure 1c-d.

This is incredibly important in quality control for many of the analyses performed in the manuscript. Strong consideration should be given to depleting *C32E8.9* by other means such as degron-based technology, availability of which is widespread. Are mutants of *C32E8.9* available? Viable?

We acknowledge the reviewer's suggestion to explore additional methods to deplete *C32E8.9* for quality control in our experiments. We generated a CRISPR-Cas9 knockout of *C32E8.9* as a robust and effective approach to eliminate its function. We successfully constructed a *C32E8.9* knockout strain, which was both viable and fertile under laboratory conditions. The detailed methodology for the design, construction, and validation of this knockout strain has been included in Methods page 16, Results page 5 and Supplementary Figure 1f.

To confirm the role of *C32E8.9* in the context of *mrps-5* RNAi, we conducted lifespan assays on *mrps-5* RNAi-treated *C32E8.9* knockout worms compared to controls. Our results demonstrated that the lifespan extension induced by *mrps-5* RNAi was completely abolished in the *C32E8.9* knockout strain, confirming our previous RNAi-based data that *C32E8.9* is required for this beneficial effect. Additionally, the knockout strain did not exhibit significant differences in baseline lifespan compared to wild-type worms, suggesting that loss of *C32E8.9* alone does not impact longevity. These findings have been added to page 6, Figure 1f.

We also assessed immune responses using a PA01 resistance assay. Consistent with the lifespan results, *mrps-5* RNAi treatment did not confer increased survival in the *C32E8.9* knockout strain, unlike in control worms. This further supports the conclusion that *C32E8.9* is required for the beneficial effects of *mrps-5* RNAi. These data are now included in Results page 9-10, Figure 5c.

In general the statistical tests (and multiple hypothesis correction) used are only spottily indicated in the figures, e.g. missing in 1e, 1g. In figure 2, the statistical test used is frankly inappropriate (t-test comparisons are not appropriate across multiple groups (ANOVA or similar should be used) and these tests require multiple hypothesis correction).

The same is true for figure 4J, 5E, 5F. This calls some of the conclusions into question and these issues must be addressed to establish validity of the results presented. In general the results should be gone through, tests should be specified, and appropriate multiple hypothesis testing correction conducted where appropriate, and t-test generally only used for data with two groups with only a pairwise comparison.

We have thoroughly reviewed and updated the statistical analyses to ensure that statistical tests are provided throughout. For comparisons involving multiple groups, we have replaced t-tests with one-way ANOVA. These revisions have been indicated in the updated figure legends. Taken together, the vast majority of the significance remained consistent with the previous analyses, and the conclusions of our study remain unchanged.

Echoing the concern above, double RNAi is not a preferred method when viable loss of function mutants are available, as in the case of *sma-3* and *sma-4*. Given that these are key conclusions made by the manuscript, loss of function mutants should be used to substantiate data with less rigorous double RNAi.

We understand the concerns regarding the limitations of using double RNAi for genetic interaction studies. We obtained the *sma-4(DR1369)* knockout strain from the Caenorhabditis Genetics Center (CGC), and performed lifespan assays to assess the effect of *mrps-5* RNAi in this strain. Consistent with our previous findings using double RNAi, we observed that *mrps-5* RNAi did not significantly extend lifespan in the *sma-4* knockout strain. These results support our original conclusion that *sma-4* is required for the beneficial effects of *mrps-5* RNAi. The data have now been included in page 8 and Supplementary Figure 2f.

It does not appear that the data substantiate the claim that “This protective effect was blunted upon double knockdown of *mrps-5* and *C32E8.9* (Fig. 5c). We observed a similar pattern with double knockdown of *mrps-5* and *sma-4* (Fig. 5d).” The relationship looks more parallel with *sma-4* than epistatic as with *C32E8.9*.

Both qPCR and RNA-seq data showed that knockdown of *C32E8.9* reverses the upregulated mRNA expression of *sma-4* induced by *mrps-5* RNAi (Supplemental Figure 1e, Supplemental Figure 2i), suggesting a potential regulatory connection. This regulation of expression does not necessarily clarify whether the interaction between *sma-4* and *C32E8.9* is epistatic or parallel in functionally contributing to the protective effect observed with *mrps-5* RNAi. We have hence toned down the conclusions on this part, and acknowledge the limitations of the current study (see page 14).

The conclusion from figure 6B that the lipidome does not change with *mrps-5* RNAi suggests that in fact remodeling of the lipidome (at least as can be assessed with static measurements such as LCMS lipidomics) is not involved in the lifespan extension of *mrps-5* knockdown. This is misleading, and perhaps the enrichment analysis done for classes of lipids as in figure 6D should be showcased first and the point made more strongly with these data.

We have adapted our initial statement from figure 6B regarding the lack of lipidome remodeling in response to *mrps-5* RNAi. We have now revised our conclusions in Figure 6b to clarify that while no significant changes in overall lipid levels were detected using static LCMS lipidomics, this does not exclude the possibility of subtle or dynamic remodeling contributing to lifespan extension (page 10 and 15).

Following your suggestion, we adjusted the flow to highlight the enrichment analysis in Figure 6d earlier in the results (page 10). This analysis suggests potential changes at the lipid class level, which may influence lifespan, even if individual lipid species do not show significant alterations.

However, the conclusion that TG, particularly those increased in chain length and desaturation, which should be decreased in abundance with activation of fatty acid catabolism, is puzzling and does not immediately follow from the data presented. The conclusion that the remodeling of the lipidome is linked to the “activation” of *C32E8.9* expression and the shift towards longer, more desaturated fatty acids could be explicitly tested with targeted experiments. The authors should address the possibility that genes encoding enzymes involved in fatty acid elongation or desaturation act mechanistically downstream of *mrps-5* and/or *C32E8.9*.

We appreciate the reviewer’s insightful comments regarding whether enzymes involved in fatty acid metabolism act downstream of *mrps-5* and *C32E8.9*. To explore this, we conducted additional analyses focusing on *fat-1* and *fat-7*, which are key regulators of fatty acid desaturation (Reviewer Figure 2a).

Our qPCR analysis revealed that *fat-1* expression was upregulated with *mrps-5* RNAi and further increased with double RNAi (*mrps-5* + *C32E8.9*). Conversely, *fat-7* expression was downregulated with *mrps-5* RNAi but increased under the double RNAi condition (see Reviewer Figure 2b-c). These results suggest that *mrps-5* RNAi, alone or combined with *C32E8.9* RNAi, affects the expression of these desaturase enzymes.

To assess if these desaturases influence the lifespan extension induced by *mrps-5* RNAi, we performed lifespan assays using *fat-1(bx24)* and *fat-6/7(bx156)* mutants. *mrps-5* RNAi was still able to significantly extend lifespan in both mutants, indicating that while *fat-1* and *fat-7* are regulated by *mrps-5*, their modulation does not appear to be directly responsible for the lifespan extension effect (Reviewer Figure 2d-e). Instead, it may reflect a broader metabolic shift associated with mitochondrial translation inhibition, involving TG remodeling and *C32E8.9* activation through pathways independent of fatty acid desaturation. Since these findings do not directly add to our conclusions, we have chosen not to include them in the manuscript.

Reviewer Figure 2 | Lifespan of *mrps-5* RNAi in fatty acid desaturase mutant worms. a | Pathway of fatty acid elongation and desaturation in *C. elegans*. **b-c |** Relative expression of *fat-1* and *fat-7* measured by qPCR in N2 worms treated with various RNAi bacteria. Statistical analysis was performed using the Analysis of Variance (ANOVA) test. Significance levels are denoted as follows: *** represents p-value < 0.001, ** represents p-value < 0.01. **d-e |** Lifespan measurements in *fat-1(bx24)* and *fat-6/7(bx156)* mutants. p-values were calculated using the log-rank test to compare each condition with the corresponding control. Statistical significance is indicated as follows: **** represents p-value < 0.0001.

Minor comments:

Figure 2 is all negative data essentially and could be re-purposed to the supplement.

While we understand the suggestion to move this data to the supplementary section, we believe it is essential to keep it in the main manuscript. Figure 2 is important for demonstrating that *C32E8.9* is specifically required for the lifespan extension induced by *mrps-5* RNAi, which is central to our hypothesis. Notably, these effects are independent of the unfolded protein response of mitochondria (UPR^{mt}), traditionally seen as key in lifespan extension. This highlights a unique, UPR^{mt} -independent role for *C32E8.9* and suggests an alternative mechanism by which mitochondrial translation inhibition can enhance healthspan. Retaining

Figure 2 in the main text emphasizes the novelty of our findings and preserves the flow of our narrative. We hope this clarifies our rationale.

There is a fair amount of negative data (e.g. 5A) presented before more meaningful data and this detracts from the message presented.

We have moved Figure 5a (the network between *sma-4*, *C32E8.9*, and *mrps-5*) to Supplementary Figure 2d to streamline the main text and focus on the key findings.

Reviewer #3 (Remarks to the Author): Inhibition of mitochondrial translation results in lifespan extension, but it is not understood the molecular basis for this increase in longevity. To address this problem, the authors analyze RNA-seq data and take a candidate approach to identify genes that alleviate the beneficial effects of knocking down *mrps-5*. Using a dual RNAi approach the authors show that knockdown of *C32E8.9* reverses many of the effects of *mrps-5* knockdown including lifespan extension, small body size, and lipidomic changes. Knockdown of *mrps-5* also causes an immune response that is absent when *C32E8.9* is concurrently knocked down. In addition, the authors find that knockdown of the transcription factor *sma-4* also reverses effects caused by *mrps-5* knockdown. Together, the authors suggest that *C32E8.9* is a necessary factor for mediating the impact of mitochondrial translation inhibition.

Main comment:

1. Almost all of the authors' results relies on observations that double RNAi knockdown of *mrps-5* with either *C32E8.9* or *sma-4* results in alleviation of various *mrps-5* phenotypes. The authors interpret these results as *C32E8.9* or *sma-4* being necessary for the effects caused by the *mrps-5* knockdown. An alternative interpretation of this data is that dual feeding of RNAi bacteria is causing less efficient knockdown of *mrps-5*. This phenomenon has been previously observed in *C. elegans* and reported in several publications (doi: 10.1186/gb-2000-2-1-research0002, DOI: 10.2144/000113365). The authors need to use an alternative to dual RNAi knockdowns to show these effects. Mutant alleles exist for *sma-4*, and although *C32E8.9* and *mrps-5* mutant alleles are not available from the strain The Caenorhabditis Genetics Center, generating mutants in *C. elegans* is now straightforward. The authors should minimally repeat the phenotypic experiments in genetic mutants and ideally would also confirm their transcriptomic and metabolomic data with these mutants.

Thank you for your thorough summary and positive feedback on our study. Below we explain how we have addressed your comments point by point.

We understand the concern about the efficiency of RNAi in dual knockdown experiments and its potential impact on our findings. To address this, we have taken extensive additional steps to validate our results using alternative approaches beyond double RNAi.

1. Validation of RNAi Efficiency:

To rule out reduced efficiency in dual RNAi, we measured *mrps-5* expression using qPCR in both single *mrps-5* RNAi and double RNAi conditions (*mrps-5* + *C32E8.9*). Our analysis showed no significant differences in *mrps-5* knockdown efficiency between the two conditions (Supplementary Figure 2c-d).

2. Construction of Genetic Mutants for Validation:

2.1 *C32E8.9* Knockout: We generated a *C32E8.9* knockout strain via CRISPR-Cas9, which was viable and fertile (page 16). Validation data are detailed in page 5 and Supplementary Figure 1f.

2.2 Lifespan Analysis: In *C32E8.9* knockout worms, the lifespan extension normally observed with *mrps-5* RNAi was abolished, confirming the necessity of *C32E8.9* for this effect. The lifespan of the *C32E8.9* knockout strain itself was comparable to wild-type controls (page 5-6, Figure 1f).

2.3 PA01 Resistance Assay: Similarly, *mrps-5* RNAi did not improve survival in the *C32E8.9* knockout strain under pathogen stress, supporting the essential role of *C32E8.9* in this context (page 9-10, Fig. 5c).

3. Validation using *sma-4* Knockout:

We also performed lifespan assays using the *sma-4* knockout strain *sma-4(DR1369)*. Consistent with our previous findings, *mrps-5* RNAi did not extend lifespan in this mutant, supporting the role of *sma-4* in mediating the effects of *mrps-5* knockdown (page 8 and Supplementary Figure 2f).

These additional validation experiments, using *C32E8.9* and *sma-4* mutants, provide strong evidence supporting our main conclusions that both genes are required for the beneficial effects of *mrps-5* RNAi. We believe this validation further strengthens the robustness of our study.

Although having genetic mutants of *C32E8.9* will be necessary to fully support the role of *C32E8.9* in suppressing the effects of mitochondrial translation inhibition, the authors could also test if *C32E8.9* RNAi knockdown by itself reverses the phenotypes caused by doxycycline inhibition of mitochondrial translation.

In our original manuscript (Figure 1d), we showed that *mrps-5* RNAi enhanced the mobility of GMC101 worms, and this effect was abolished when combined with *C32E8.9* RNAi. To answer the reviewer's question, we tested whether *C32E8.9* RNAi alone could impact the beneficial effects induced by doxycycline treatment. Upon knocking down *C32E8.9* in doxycycline-treated GMC101 worms, we observed that the enhanced mobility returned to control levels. These new results are included in page 5 and Supplementary Figure 1g, with confirmation of *C32E8.9* knockdown efficiency in Supplementary Figure 1h.

In summary, this additional experiment supports our conclusion that *C32E8.9* plays a crucial role in mediating the beneficial effects of mitochondrial translation inhibition.

Minor comment:

1. In Figure 1b there are no units to the scale.

Thank you for pointing this out. We have updated Figure 1b to include separate scales for mRNA and protein in *C. elegans* as well as mRNA in mice, providing better clarity and ensuring consistent data representation.

Reviewer #1:

The authors have effectively addressed all my concerns, and the manuscript is suitable for publication in Nature Communications after the following minor textual corrections are made:

Thank you for your positive assessment and thoughtful comments. We have addressed the suggestions as detailed below.

1. Line 66, page 3: *C. elegans* should be italicized.

We have carefully reviewed the manuscript and ensured that *C. elegans* is consistently italicized throughout the text.

2. Lines 145–146, page 6: Supplementary Fig. 1f should be cited to demonstrate the efficiency of the *C32E8.9* knockout.

We have now added a citation to Supplementary Fig. 1f at the appropriate location to clearly demonstrate the efficiency of the *C32E8.9* knockout.

3. In the legend of Supplementary Fig. 3i, please include the genotype of the *C32E8.9* overexpression strain to help readers understand the source of the fluorescent signals.

We have included the genotype of the *C32E8.9* overexpression strain in the figure legend for clarity.

4. Please provide a brief explanation for the use of the *dpy-30* promoter to overexpress *C32E8.9* and clarify why the GFP signal is predominantly observed in the pharynx. Could the strong GFP signal result from leakage from the strong *Pmyo-2::mCherry*?

These minor revisions will enhance the clarity and completeness of the manuscript.

We selected the *dpy-30* promoter because it is ubiquitously expressed, and little is known about the tissue-specific expression of *C32E8.9* currently. Using a broadly expressed promoter allowed us to examine potential expression patterns without introducing bias. We have added a brief explanation in the Methods section to clarify this choice (Line 531-533).

To better illustrate GFP distribution, we have included representative higher-magnification images in Supplementary Fig. 4. In the *C32E8.9* overexpression strain, GFP was detected throughout the body, while mCherry expression was restricted to the pharynx. The stronger GFP signal observed in the pharynx is likely due to spectral bleed-through between the GFP and mCherry channels. When mCherry expression under the *myo-2* promoter is particularly strong, its fluorescence can be detected in the GFP channel, leading to an apparent increase in GFP signal in the pharynx. We have included an explanation of this observation for clarity. (Line 338 to 341, line 542)

Reviewer #2:

The authors have done a suitable job of addressing my concerns, most of which were methodological and statistical. The manuscript is interesting and adds new dimensions to our understanding of stress responses in aging and will be of interest to a broad readership.

Thank you for your thoughtful review and positive feedback. We are glad that you find the manuscript interesting and valuable for advancing the understanding of stress responses in aging. Your insights have helped improve the clarity and rigor of our work.

Reviewer #3:

The inclusion of the *C32E8.9* mutant data greatly strengthens the manuscript, but although the authors mostly address my concerns, there are still several issues that should be addressed.

Thank you for your careful review and thoughtful feedback. We have addressed your comments point by point.

Major points

Comment: 1. Although the inclusion of mutants now alleviates the concern that the phenotypes observed were due to a reduction in RNAi efficiency in the double knockdown condition, the data included by the authors in S1C, appears to show a reduction of *C32E8.9* RNAi efficiency, albeit not significant to $p < .05$. I feel that authors should mention this as a potential caveat to their study in the discussion.

We appreciate this observation. While the reduction in *C32E8.9* RNAi efficiency in the double knockdown condition in Figure S1C is not statistically significant, we acknowledge that it may still have biological relevance. We have added a statement in the discussion to acknowledge this as a potential caveat in our study (Line 483 to 489).

2. The authors state in their reporting summary that all data was replicated at least twice, but table s1 shows that several of the lifespans in figure 4 only have a single replicate. Most of the figures legends do not say how many animals were used for each experiments and how many biological replicates there were. This information needs to be included in the figure legends and in the reporting summary. The individual points should be shown for the survival curves. All the survival curve data should be shown, either on the same graph, or in the supplementary.

We confirm that all lifespan assays presented in this manuscript were replicated at least twice, where key experiments always consisted of $n \geq 3$ replicates. The lifespan data in Figure 4 represent RNAi/knock-out screening for candidate genes, for which we initially did not provide details of the replicates in the supplementary material, as the screening did not alter the main conclusions of the study. We have now included all replicates for completeness. Regarding the survival data, we note that no smoothing was applied to the lifespan curves, meaning that each of the bends in the curve represents an actual measurement day, from which individual data points can be inferred. To enhance clarity, we have also updated the figure legends to specify the number of biological replicates and animals used in each experiment. Additionally, all survival curve data, along with summarized statistics, are now included in Supplementary Table S1.

3. The data availability statement for the paper says there is a source data file, but that does not appear to have been included or not made available to the reviewers.

The source data of transcriptome in this study have been deposited in the GEO database under the accession number GSE248642. Reviewers can access the data using the secure

token "crmfkomcplsrby" while it remains in private status. The lipidomics data generated in this study are provided in the Source Data file accompanying this paper. (Mentioned in Line 690 to 691)

Minor points

1. Figure S1. The graphs are missing units. Several of the graphs list "dox" as an RNAi condition. I believe it should be ctl +dox.

We have reviewed and corrected the figure labels to ensure clarity. Units have been added where missing, and "dox" has been updated to "control + dox" as suggested.

2. Figure 2d. Scale bars should be included.

We have added scale bars to Figure 2d for clarity.

Reviewer #1 (Remarks to the Author):

The authors have addressed all my remaining concerns.

We sincerely thank the reviewer for the positive feedback and are glad that all remaining concerns have been addressed to your satisfaction. We truly appreciate your time, thoughtful input, and support throughout the review process.

Reviewer #3 (Remarks to the Author):

The authors have addressed all my concerns and I now enthusiastically support publication of their manuscript

We greatly appreciate your thoughtful review and are delighted that you now enthusiastically support the publication of our manuscript. Thank you for your valuable feedback and continued support during the entire review process.